# Neutrophil-generated HOCl leads to non-specific thiol oxidation in phagocytized bacteria

Adriana Degrossoli[1†], Alexandra Müller[1], Kaibo Xie[1], Jannis F Schneider[1], Verian Bader[2], Konstanze F Winklhofer[2], Andreas J Meyer[3], Lars I Leichert[1]*

[1]Institute for Biochemistry and Pathobiochemistry – Microbial Biochemistry, Ruhr-Universität Bochum, Bochum, Germany; [2]Institute for Biochemistry and Pathobiochemistry – Molecular Cell Biology, Ruhr-Universität Bochum, Bochum, Germany; [3]INRES – Chemical Signalling, Rheinische Friedrich-Wilhelms-Universität Bonn, Bonn, Germany

*For correspondence:
lars.leichert@ruhr-uni-bochum.de

Present address:
†Departamento de Ciências da Saúde, Universidade Federal de Lavras, Lavras, Brazil

Competing interests: The authors declare that no competing interests exist.

Reviewing editor: Christine Winterbourn,

**Abstract** Phagocytic immune cells kill pathogens in the phagolysosomal compartment with a cocktail of antimicrobial agents. Chief among them are reactive species produced in the so-called oxidative burst. Here, we show that bacteria exposed to a neutrophil-like cell line experience a rapid and massive oxidation of cytosolic thiols. Using roGFP2-based fusion probes, we could show that this massive breakdown of the thiol redox homeostasis was dependent on phagocytosis, presence of NADPH oxidase and ultimately myeloperoxidase. Interestingly, the redox-mediated fluorescence change in bacteria expressing a glutathione-specific Grx1-roGFP2 fusion protein or an unfused roGFP2 showed highly similar reaction kinetics to the ones observed with roGFP2-Orp1, under all conditions tested. We recently observed such an indiscriminate oxidation of roGFP2-based fusion probes by HOCl with fast kinetics in vitro. In line with these observations, abating HOCl production in immune cells with a myeloperoxidase inhibitor significantly attenuated the oxidation of all three probes in bacteria.

DOI: https://doi.org/10.7554/eLife.32288.001

## Introduction

When bacteria encounter professional phagocytic immune cells, such as neutrophils or macrophages, they are engulfed and phagocytized. Within the phagosome, an intracellular compartment formed during phagocytosis, bacteria are exposed to a complex mixture of toxins, chiefly among them oxidative and nitrosative species (*Hurst, 2012*; *Segal, 2005*; *Urban et al., 2006*; *Winterbourn and Kettle, 2013*). This mechanism, termed respiratory burst, is initiated by the reduction of oxygen to superoxide radicals through NADPH oxidase NOX2, an enzyme that is assembled at the phagosomal membrane (*Chanock et al., 1994*; *Segal and Abo, 1993*). From superoxide, other reactive oxygen species (ROS), such as hydrogen peroxide ($H_2O_2$), are formed and released into the phagosomal space (*Hampton et al., 1998*). The respiratory burst is potentiated through lysosomal degranulation that releases myeloperoxidase (MPO) and other microbicidal proteins into the phagosome (*Hurst, 2012*). MPO catalyzes the formation of hypochlorous acid (HOCl), a strong oxidant, from $H_2O_2$ and chloride ions (*Furtmüller et al., 2003*). These ROS can oxidize and damage virtually any cellular molecule, and together with other subsequent mechanisms, ultimately lead to microbial death. In contrast, individuals with chronic granulomatous disease (CGD), a genetic disease with impaired NADPH oxidase activity, as well as mice that lack components of the NADPH oxidase are strongly susceptible to microbial infection (*Hampton et al., 1998*; *Holmes et al., 1967*; *Mandell, 1974*; *Segal, 2005*; *Segal et al., 2000*; *Vethanayagam et al., 2011*; *Winkelstein et al., 2000*).

**eLife digest** A group of cells of the immune system defends the body against infections by wrapping themselves around bacteria, and effectively 'eating' them. During this process, called phagocytosis, the cell also douses the bacterium with a deadly cocktail of chemicals, including an antiseptic – hydrogen peroxide – and bleach. This mixture chemically burns, and then kills, the invader. The immune cells create hydrogen peroxide and bleach through chemical reactions that require two enzymes, NOX2 and MPO. The NOX2 enzyme is activated first, and produces a compound which is then transformed into hydrogen peroxide. In turn, hydrogen peroxide is used by MPO to make bleach.

Phagocytosis is still poorly understood, and difficult to study: for example, it is not clear when the toxic mix is released, and which of its components are the most important. Here, Degrossoli et al. peer into this process: to do so, they genetically engineer bacteria and give them a built-in chemical burn tracker. The bacteria are made to carry fluorescent proteins which normally glow under blue light, but start to also react to violet light if they are exposed to a chemical burn.

Under the microscope, when these bacteria encounter immune cells, they start glowing under violet light only a few seconds after they have been phagocytized. This shows that, during phagocytosis, the chemical mix is used almost immediately. The new technique also reveals that cells without a working NOX2 enzyme – which cannot produce hydrogen peroxide – could not burn the bacteria. However, hydrogen peroxide is also used by MPO to create bleach. If just MPO is deactivated, the cells can burn the bacteria, but much less efficiently. This, and the speed with which these fluorescent proteins were burnt, shows that the bleach is the main component of the toxic mix used during phagocytosis.

Chronic granulomatous disease is a condition where patients can have a faulty version of NOX2, which makes it harder for them to fight infection. Understanding the mechanisms and the enzymes associated with phagocytosis could lead to improved treatment in the future.

DOI: https://doi.org/10.7554/eLife.32288.002

The mechanisms that kill bacteria in phagocytic immune cells are still not fully understood. However, investigation of the phagosomal environment is quite challenging due to its transient nature and the complexity of the mixture of oxidative species (*Nüsse, 2011*).

Several methods have been employed for the detection of phagosomal ROS, including fluorescent redox-sensitive dyes. However, most of these methods have a number of limitations such as non-specificity, irreversibility, non-quantitative information, or a lack of subcellular localization (*Nauseef, 2014*). The most widespread compound to detect $H_2O_2$ in intact cells is 2',7'-dihydrodichlorofluorescein ($H_2DCF$), which can be oxidized to fluorescent 2',7'-dichlorofluorescein (DCF) (*Chen et al., 2010*; *Maghzal et al., 2012*). DCF oxidation, however, is considered mainly qualitative, as it is observable in the absence of $H_2O_2$ and is stimulated by metals, peroxidases, and cytochrome c. Therefore, it does not provide detailed quantitative and compartment-specific information (*Meyer and Dick, 2010*; *Rota et al., 1999*; *Tarpey et al., 2004*). Recently, roGFP2 (reduction-oxidation-sensitive green fluorescent protein 2) has been used to study oxidative and nitrosative stress dynamics in *Salmonella* inside macrophages (*van der Heijden et al., 2015*). roGFP2 has several advantages when compared to commercially available fluorescent redox-sensitive dyes. As a GFP variant, it can be genetically introduced into virtually any biological system and can be even targeted to specific cellular compartments (*Dooley et al., 2004*; *Hanson et al., 2004*). Its redox state, which depends on the redox state of the biological system, can then be measured with the help of an engineered pair of cysteine residues close to the fluorophore. The reversible disulfide bond formation between these cysteines triggers a slight conformational change, which results in a reversible change of the protonation status of the fluorophore. The reduced and oxidized form of roGFP2 therefore have distinct fluorescence excitation maxima at 395 and 490 nm, respectively (*Dooley et al., 2004*). Either the 405/488 nm ratio with laser-based excitation or 390/480 nm ratio on filter-based recording devices can thus be used to directly determine the probe's redox state (*Meyer and Dick, 2010*). This ratiometric approach compensates for variations due to differences in absolute roGFP2 concentrations, allowing for quantitative monitoring. These probes thus allow compartment-specific real-

time ratiometric quantification of the intracellular redox status in prokaryotic as well as eukaryotic cells (*Arias-Barreiro et al., 2010*; *Bhaskar et al., 2014*; *Meyer and Dick, 2010*; *van der Heijden et al., 2015*).

Here, we report the use of three different roGFP2-based fluorescent redox probes to quantitatively track the redox state of bacteria during the phagocytic process. Using the $H_2O_2$-sensitive roGFP2-Orp1 probe expressed in the cytoplasm of *Escherichia coli*, we could show with fluorescence spectroscopy and quantitative fluorescence microscopy that phagocytosis by a neutrophil-like cell line leads to probe oxidation within seconds. Comparison of roGFP2-Orp1's oxidation kinetics to the oxidation kinetics of the glutathione-specific Grx1-roGFP2 probe and the oxidation kinetics of unfused roGFP2 suggested that the presence of a strong oxidant in the phagosome is over-riding the specificity of the fusion probes. Based on previous in vitro studies and chemical inhibition of myeloperoxidase, we conclude that HOCl is the major reactive species during the onset of the respiratory burst in neutrophils and initiates non-specific thiol oxidation in phagocytized bacteria.

## Results

### Expression of the roGFP2-based fusion probes in *Escherichia coli*

When bacteria are phagocytized by professional phagocytic immune cells, they are exposed to a toxic cocktail of reactive oxygen and nitrogen species, including hydrogen peroxide ($H_2O_2$) generated by the action of NADPH-oxidase and superoxide dismutase. We wanted to monitor the increase in $H_2O_2$ and potentially other reactive species inside the bacterial cell during those host-pathogen interactions in situ. Thus, we expressed the roGFP2 fusion probe roGFP2-Orp1 in *Escherichia coli* MG1655. This probe is specifically designed to measure $H_2O_2$ in biological systems.

We could express roGFP2-Orp1 stably in *E. coli* from a plasmid (*Figure 1A*). Using fluorescence spectroscopy, we could determine the oxidation state of the probe inside the *E. coli* cytoplasm using the ratio between the excitation wavelengths of 405 and 488 nm (*Dooley et al., 2004*; *Gutscher et al., 2008*; *Hanson et al., 2004*). Addition of the strong oxidant Aldrithiol-2 (AT-2, 2,2′-Dipyridyl disulfide) to the bacterial cells led to full oxidation of the probe, while addition of DTT resulted in full reduction (*Figure 1D and G*). The exposure to reactive species in the phagolysosome could also interfere with the glutathione redox potential ($E_{GSH}$) within the cell. Thus, we introduced an expression plasmid encoding Grx1-roGFP2 into *E. coli*. Grx1-roGFP2 was specifically designed to measure the glutathione redox potential ($E_{GSH}$).We could stably express Grx1-roGFP2 in *E. coli* (*Figure 1B*), fully reduce it with DTT, and fully oxidize it with AT-2 (*Figure 1E and H*). Additionally, we expressed unfused roGFP2 in *E. coli* and performed the same experiments (*Figure 1C,F,I*).

### roGFP2-Orp1 expressed in *E. coli* reacts toward exogenous hydrogen peroxide

Having established that the probe could be fully reduced and fully oxidized in *E. coli*, we then tested the response of the roGFP2-Orp1 probe towards exogenous $H_2O_2$. As expected, roGFP2-Orp1 changed its redox state when *E. coli* was exposed to a bolus of exogenous hydrogen peroxide at concentrations as low as 10 μM (*Figure 1J*). Within a short period of time, the probe's redox state returned to the pre-$H_2O_2$ steady state, indicating that the bacterial cell can recover from the oxidative insult. Higher concentrations of exogenous hydrogen peroxide led to longer recovery times of the probe, until at concentrations of 10 mM, the probe did no longer recover (*Figure 1J*).

### PLB-985 cells, when differentiated to neutrophils, phagocytize *E. coli*

Neutrophil granulocytes are the first line of defense against bacteria in the blood stream. They can phagocytize microorganisms and kill them. One of the main killing factors employed by neutrophils is the production of reactive oxygen and nitrogen species during the so-called oxidative burst (*Segal, 2005*; *Verchier et al., 2007*). For our experiments, we selected the myeloid PLB-985 cell line that can be differentiated to neutrophil-like cells (*Pivot-Pajot et al., 2010*). To differentiate PLB-985 cells we exposed them to DMSO (*Pivot-Pajot et al., 2010*) and added interferon γ (IFNγ). After 5 days of incubation, differentiated cells showed significantly higher expression of CD11b and CD64 than undifferentiated cells, while CD16 and CD66b expression was unaltered (*Figure 2A–D*). Differentiated cells lost their spherical shape, and the cells showed a morphology typical of neutrophils

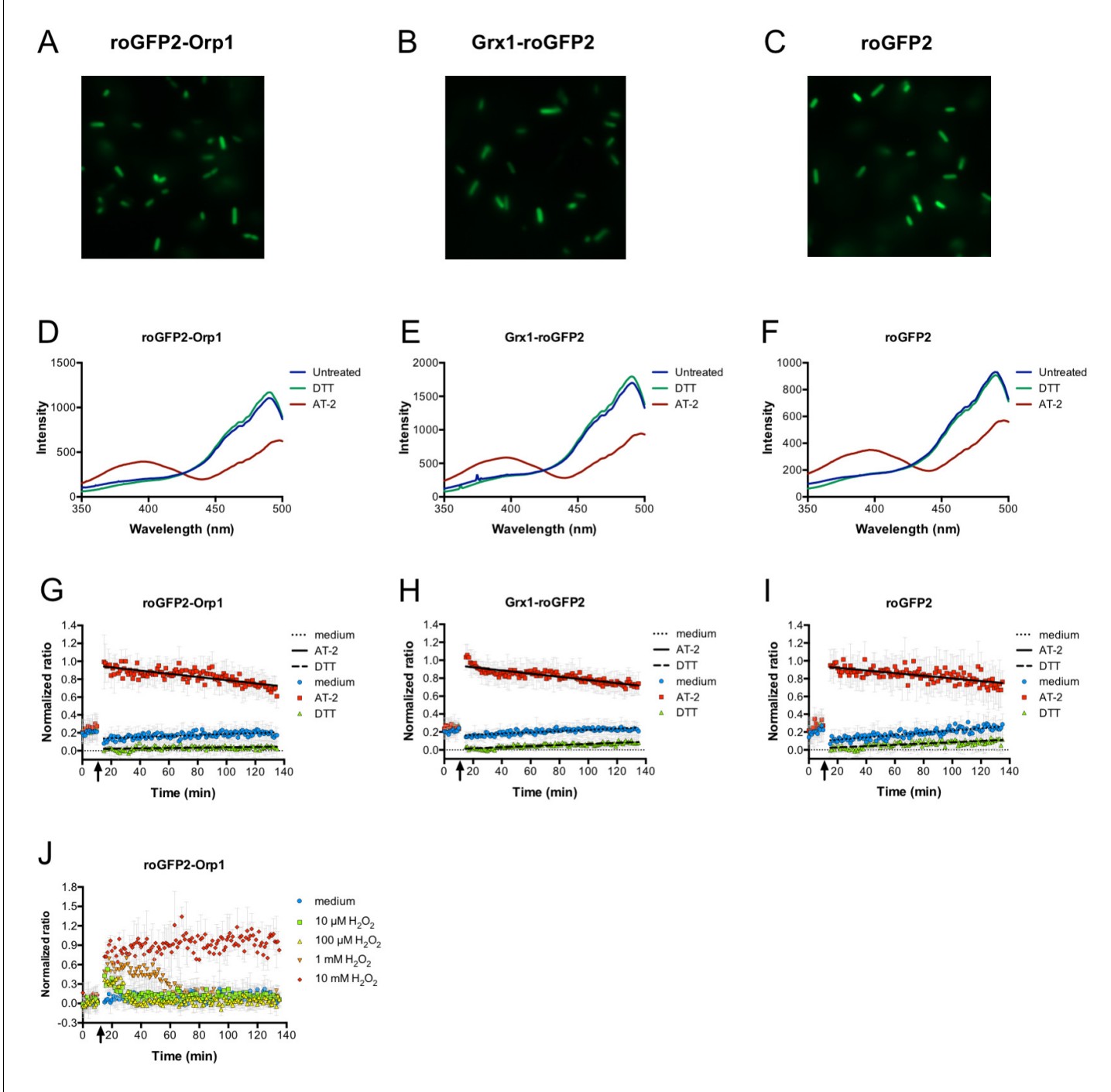

**Figure 1.** Expression of genetically encoded redox probes in *E. coli*. Fluorescence microscopy reveals uniform expression of roGFP2-based probes (fused to Orp1 and Grx1 and unfused probe) from a plasmid in the cytoplasm of *E. coli* (**A–C**). The probes' response to the strong thiol reductant DTT and the strong thiol oxidant AT-2 could be measured in a fluorescence spectrophotometer by monitoring the characteristic excitation spectra (**D–F**). A normalized ratio of the intensity of fluorescence at 405 and 488 nm excitation allowed for a time-course measurement of the probes' oxidation in response to DTT and AT-2 treatment in a 96-well plate reader. The arrows indicate the time point of addition of the oxidant and reductant. Medium served as a control (**G–I**). The level of oxidation caused by hydrogen peroxide in the hydrogen peroxide-sensitive probe roGFP2-Orp1 expressed in *E. coli* can be followed in a plate reader. The arrow indicates the addition of hydrogen peroxide at the concentrations indicated (**J**).
DOI: https://doi.org/10.7554/eLife.32288.003

The following source data is available for figure 1:

**Source data 1.** Numerical fluorescence spectrometry data represented in *Figure 1D*.

*Figure 1 continued on next page*

*Figure 1 continued*

DOI: https://doi.org/10.7554/eLife.32288.004

**Source data 2.** Numerical fluorescence spectrometry data represented in *Figure 1E*.

DOI: https://doi.org/10.7554/eLife.32288.005

**Source data 3.** Numerical fluorescence spectrometry data represented in *Figure 1F*.

DOI: https://doi.org/10.7554/eLife.32288.006

**Source data 4.** Numerical fluorescence plate reader data represented in *Figure 1G*.

DOI: https://doi.org/10.7554/eLife.32288.007

**Source data 5.** Numerical fluorescence plate reader data represented in *Figure 1H*.

DOI: https://doi.org/10.7554/eLife.32288.008

**Source data 6.** Numerical fluorescence plate reader data represented in *Figure 1I*.

DOI: https://doi.org/10.7554/eLife.32288.009

**Source data 7.** Numerical fluorescence plate reader data represented in *Figure 1J*.

DOI: https://doi.org/10.7554/eLife.32288.010

when stained with May-Grünwald-Giemsa stain (*Figure 2E–H*). Differentiated cells internalized IgG-opsonized *E. coli*, demonstrating their phagocytic capacity (*Figure 3A*). In contrast, the usage of undifferentiated PLB-985 cells or non-opsonized *E. coli* led to significantly decreased internalization of bacteria, despite their presence in the surrounding medium (*Figure 3B and D*). When IFNγ was not added during the differentiation process, phagocytosis was less pronounced (*Figure 3C*). Our microscopic data was corroborated and quantified by flow cytometry (*Figure 3G–I*). For subsequent experiments, we used DMSO+ IFNγ-differentiated cells and opsonized *E. coli*, because under those conditions the neutrophil-like cells had the highest phagocytic capacity.

## roGFP2-based probes are oxidized in phagocytized *E. coli*

We then monitored the oxidation state of roGFP2-Orp1 in *E. coli* during phagocytosis. Thus, roGFP2-Orp1-expressing *E. coli* were co-incubated with differentiated neutrophil-like PLB-985 cells and the ratio of fluorescent intensity at 405/488 nm excitation was measured as readout of the probe's oxidation state in a fluorescence plate reader. The oxidation state of the probe increased, when opsonized *E. coli* expressing roGFP2-Orp1 were incubated with neutrophil-like cells. The oxidation kinetic showed a gradual increase in the 405/488 nm ratio, as soon as *E. coli* was incubated with neutrophil-like PLB-985, reaching a plateau after 60 min of incubation and remaining stably oxidized until the end of the measurement (2 hr). In contrast, the probe was not oxidized when bacteria were incubated with medium or undifferentiated cells (*Figure 4A*). When *E. coli* expressing Grx1-roGFP2 was cultured with neutrophil-like PLB-985 cells, the redox response of the probe was highly similar to the response observed with roGFP2-Orp1 (*Figure 4B*), and we essentially observed the same oxidation kinetics with the unfused roGFP2 probe (*Figure 4C*).

## Probe oxidation is dependent on effective phagocytosis

The oxidation kinetics of the roGFP2-Orp1 probe in *E. coli* cultured with differentiated PLB-985 cells was not as fast as in an *E. coli* bacteria suspension that was directly exposed to high concentrations of $H_2O_2$ (compare *Figure 1J* and *Figure 4A*). We argued that this could mean that bacteria experience redox stress from the host cells only upon phagocytosis. To examine if phagocytosis indeed plays a role in probe oxidation in the *E. coli* cytoplasm, we blocked phagocytosis by treating the neutrophil-like cells with Cytochalasin D (CD). Cytochalasin D-pre-treated, differentiated PLB-985 cells did phagocytize opsonized *E. coli* less effectively than untreated, differentiated PLB-985 cells or cells treated with DMSO (vehicle control) (*Figure 5A*). To verify the effective blocking of phagocytosis, we used opsonized FITC-labeled Zymosan and tested the phagocytic capacity of Cytochalasin D-treated and untreated PLB-985 cells. In this case, we quenched the fluorescence of attached Zymosan particles, giving us a more exact readout of the phagocytic capacity. Zymosan phagocytosis was almost completely blocked in Cytochalasin D-treated neutrophil-like cells when compared to the control or cells pre-incubated with DMSO (vehicle control) (*Figure 5B*). Having established the inhibition of phagocytosis, we tested the redox state of roGFP2-Orp1 in *E. coli* in the presence of differentiated, Cytochalasin D-treated PLB-985 cells. The 405/488 nm ratio of roGFP2-Orp1 did not change significantly over time (*Figure 5C*). Similarly, oxidation of the Grx1-roGFP2 fusion and

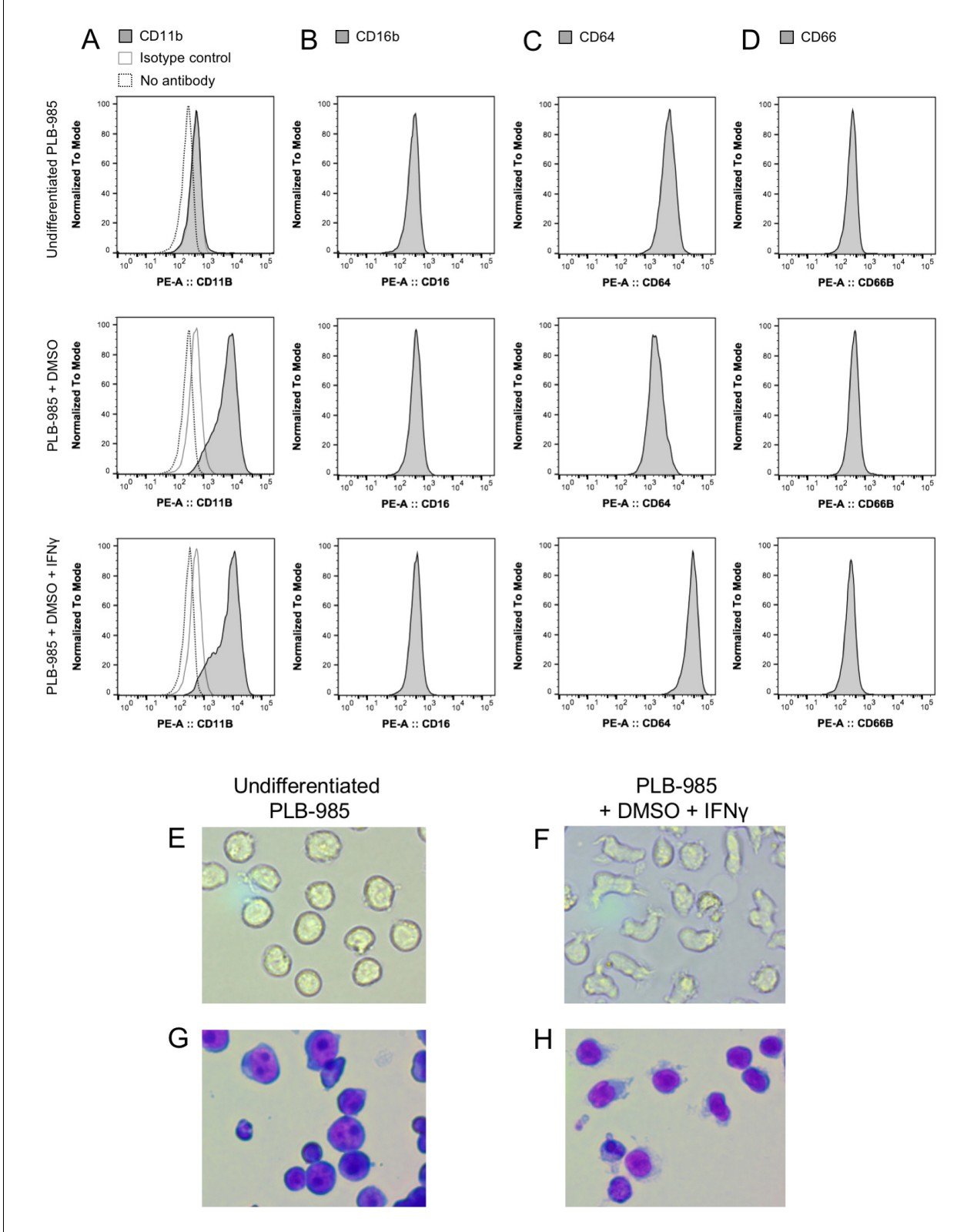

**Figure 2.** Differentiation of the myeloid cell line PLB-985 to neutrophil-like cells. Determination of surface markers CD11b, CD16b, CD64, and CD66 before and after two different differentiation protocols (A–D). Cells differentiated with DMSO or DMSO and IFNγ expressed significantly higher levels of the neutrophil markers CD11b and CD64 when compared to undifferentiated cells. These changes in the expression of surface markers coincided with morphological changes typical for neutrophils when observed under phase contrast (E–F) or stained with May-Grünwald-Giemsa stain (G–H).
*Figure 2 continued on next page*

*Figure 2 continued*

DOI: https://doi.org/10.7554/eLife.32288.011

The following source data is available for figure 2:

**Source data 1.** Numerical flow cytometry data represented in *Figure 2A*, trace CD11b, Undifferentiated.
DOI: https://doi.org/10.7554/eLife.32288.012
**Source data 2.** Numerical flow cytometry data represented in *Figure 2A*, trace CD11b, Undifferentiated, isotype control.
DOI: https://doi.org/10.7554/eLife.32288.013
**Source data 3.** Numerical flow cytometry data represented in *Figure 2A*, trace CD11b, Undifferentiated, negative control.
DOI: https://doi.org/10.7554/eLife.32288.014
**Source data 4.** Numerical flow cytometry data represented in *Figure 2A*, trace CD11b, DMSO.
DOI: https://doi.org/10.7554/eLife.32288.015
**Source data 5.** Numerical flow cytometry data represented in *Figure 2A*, trace CD11b, DMSO, isotype control.
DOI: https://doi.org/10.7554/eLife.32288.016
**Source data 6.** Numerical flow cytometry data represented in *Figure 2A*, trace CD11b, DMSO, negative control.
DOI: https://doi.org/10.7554/eLife.32288.017
**Source data 7.** Numerical flow cytometry data represented in *Figure 2A*, trace CD11b, DMSO+ IFNγ.
DOI: https://doi.org/10.7554/eLife.32288.018
**Source data 8.** Numerical flow cytometry data represented in *Figure 2A*, trace CD11b, DMSO+ IFNγ, isotype control.
DOI: https://doi.org/10.7554/eLife.32288.019
**Source data 9.** Numerical flow cytometry data represented in *Figure 2A*, trace CD11b, DMSO+ IFNγ, negative control.
DOI: https://doi.org/10.7554/eLife.32288.020
**Source data 10.** Numerical flow cytometry data represented in *Figure 2B*, trace CD16, Undifferentiated.
DOI: https://doi.org/10.7554/eLife.32288.021
**Source data 11.** Numerical flow cytometry data represented in *Figure 2B*, trace CD16, DMSO.
DOI: https://doi.org/10.7554/eLife.32288.022
**Source data 12.** Numerical flow cytometry data represented in *Figure 2B*, trace CD16, DMSO+ IFNγ.
DOI: https://doi.org/10.7554/eLife.32288.023
**Source data 13.** Numerical flow cytometry data represented in *Figure 2C*, trace CD64, Undifferentiated.
DOI: https://doi.org/10.7554/eLife.32288.024
**Source data 14.** Numerical flow cytometry data represented in *Figure 2C*, trace CD64, DMSO.
DOI: https://doi.org/10.7554/eLife.32288.025
**Source data 15.** Numerical flow cytometry data represented in *Figure 2C*, trace CD64, DMSO+ IFNγ.
DOI: https://doi.org/10.7554/eLife.32288.026
**Source data 16.** Numerical flow cytometry data represented in *Figure 2D*, trace CD66b, Undifferentiated.
DOI: https://doi.org/10.7554/eLife.32288.027
**Source data 17.** Numerical flow cytometry data represented in *Figure 2D*, trace CD66b, DMSO.
DOI: https://doi.org/10.7554/eLife.32288.028
**Source data 18.** Numerical flow cytometry data represented in *Figure 2D*, trace CD66b, DMSO+ IFNγ.
DOI: https://doi.org/10.7554/eLife.32288.029

unfused roGFP2 probe was phagocytosis-dependent (*Figure 5D & E*) indicating that bacteria are exposed to reactive oxygen species only when internalized.

## Once phagocytized, the redox state of the bacterial cytoplasm changes promptly

Our data with exogenous hydrogen peroxide demonstrated that the roGFP2-Orp1 probe reacts with very fast kinetics toward hydrogen peroxide. In contrast, the probe's 405/488 nm ratio in *E. coli* cells exposed to neutrophil-like cells changed more gradually. In combination with the significantly lower probe oxidation in *E. coli* exposed to phagocytosis-impaired neutrophil-like cells, we hypothesized that phagocytosis is the rate-limiting step for probe oxidation. To test if the probes are indeed oxidized only upon phagocytosis, we monitored the dynamics of probe oxidation using quantitative fluorescence microscopy. Our experiments show that the probe remains in a reduced state in bacteria not yet phagocytized, but this changes promptly within seconds upon phagocytosis (*Figure 6*, *Video 1*). The gradual increase in probe oxidation in *E. coli* exposed to neutrophil-like cells thus reflects the gradual phagocytosis of individual bacteria over time.

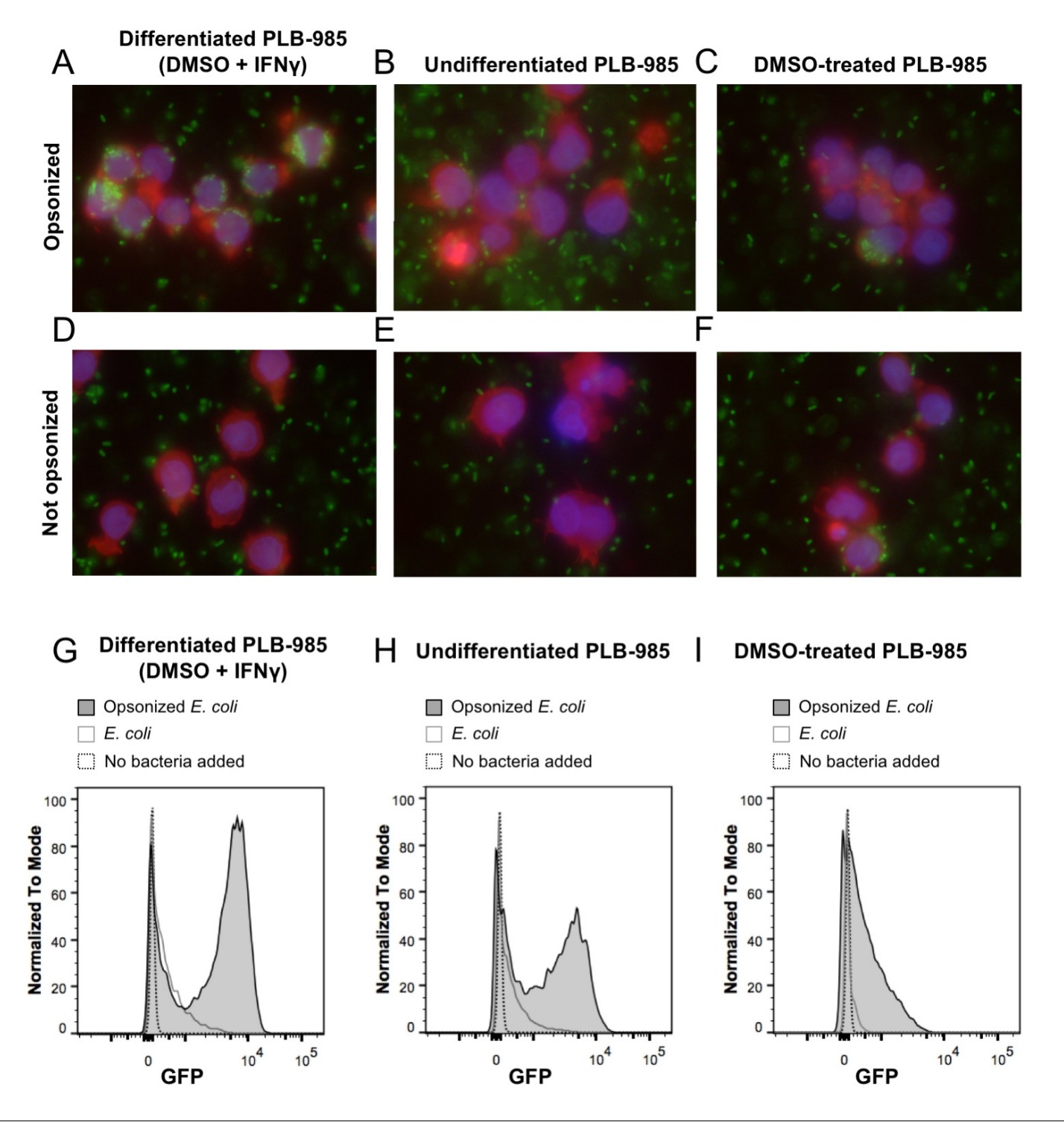

**Figure 3.** Phagocytosis of *E. coli* by PLB-985. (**A**) PLB-985 cells differentiated with DMSO and IFNγ effectively phagocytize *E. coli* opsonized with human immunoglobulin G (IgG). *E. coli* cells expressing roGFP2-Orp1 associate with the neutrophil-like cells. This is not the case with undifferentiated cells (**B**). PLB-985 cells treated with DMSO are less effective in their phagocytosis as well (**C**). Effective phagocytosis is dependent on opsonization (**D–F**). Microscopic images are composite overlays of the DAPI nuclear stain (blue), the TRITC-conjugated phalloidin-based actin stain (red) and the GFP channel (bacteria, green). Flow cytometry corroborates the microscopic evidence. Differentiated cells co-incubated with opsonized *E. coli* show the highest population of cells incorporating bacteria as measured by GFP-fluorescence (**G**), whereas the bacteria-incorporating population size in undifferentiated (**H**) and DMSO-treated cells (**I**) is significantly smaller.

*Figure 3 continued on next page*

*Figure 3 continued*

DOI: https://doi.org/10.7554/eLife.32288.030

The following source data is available for figure 3:

**Source data 1.** Numerical flow cytometry data represented in *Figure 3G*, trace Opsonized *E. coli*.
DOI: https://doi.org/10.7554/eLife.32288.031
**Source data 2.** Numerical flow cytometry data represented in *Figure 3G*, trace *E. coli*.
DOI: https://doi.org/10.7554/eLife.32288.032
**Source data 3.** Numerical flow cytometry data represented in *Figure 3G*, trace No bacteria added.
DOI: https://doi.org/10.7554/eLife.32288.033
**Source data 4.** Numerical flow cytometry data represented in *Figure 3H*, trace Opsonized *E. coli*.
DOI: https://doi.org/10.7554/eLife.32288.034
**Source data 5.** Numerical flow cytometry data represented in *Figure 3H*, trace *E. coli*.
DOI: https://doi.org/10.7554/eLife.32288.035
**Source data 6.** Numerical flow cytometry data represented in *Figure 3H*, trace no bacteria added.
DOI: https://doi.org/10.7554/eLife.32288.036
**Source data 7.** Numerical flow cytometry data represented in *Figure 3I*, trace Opsonized *E. coli*.
DOI: https://doi.org/10.7554/eLife.32288.037
**Source data 8.** Numerical flow cytometry data represented in *Figure 3I*, trace *E. coli*.
DOI: https://doi.org/10.7554/eLife.32288.038
**Source data 9.** Numerical flow cytometry data represented in *Figure 3I*, trace no bacteria added.
DOI: https://doi.org/10.7554/eLife.32288.039

## roGFP2-Orp1 oxidation is dependent on NOX2 activity

The initial superoxide generated during the oxidative burst is produced by NADPH oxidase. Superoxide itself is highly reactive, but also the originator of further reactive species that are produced subsequently during the oxidative burst, chiefly among them $H_2O_2$, through superoxide dismutase action (*Dupré-Crochet et al., 2013*; *Klebanoff et al., 2013*; *Segal et al., 1980*; *Segal, 2005*; *Winterbourn, 2014*). Thus, we tested the influence of NOX2 on probe oxidation in *E. coli*. The generation of reactive species by PLB-985 cells was confirmed using the oxidant-sensitive dye 2',7'-Dichlordihydrofluorescein-diacetate ($H_2$DCFDA), which is oxidized intracellularly to 2',7'-Dichlorfluorescein (DCF) by a number of reactive oxygen species (*Chen et al., 2010*; *Yazdani, 2015*). Neutrophil-like PLB-985 cells, when stimulated with PMA, showed higher DCF fluorescence than non-stimulated cells (*Figure 7A*), confirming previous reports that these cells do indeed produce ROS (*Segal et al., 1980*). ROS production was also stimulated by opsonized *E. coli* (*Figure 7B*). Based on the previous findings and our own data, we argued that cells lacking NOX2 activity should not be able to induce roGFP2-Orp1 probe oxidation in *E. coli* cells. PLB-985 cells lacking gp91$^{phox}$, the catalytic subunit of NOX2, were indeed unable to induce significant oxidation of roGFP2-Orp1 in *E. coli* (*Figure 8*).

## Probe oxidation is dependent on myeloperoxidase

Although PLB-985 cells lacking NOX2 activity were incapable to induce significant oxidation of roGFP2-Orp1 in *E. coli*, the virtually identical behavior of all three roGFP2-based probes in our phagocytosis experiments is inconsistent with superoxide-derived $H_2O_2$ as the main factor in probe oxidation. All three probes have different specificities: Unfused roGFP2, for example, is known to react slowly with $H_2O_2$ and glutathione in vitro and thus is not well-suited for the detection of transient and weak oxidative stress (*Meyer and Dick, 2010*). Additionally, the Grx1-roGFP2 probe rapidly equilibrates with the cell's glutathione redox potential, but is not directly affected by hydrogen peroxide (*Gutscher et al., 2008*). However, we showed recently, that, in vitro, the specificity of these three probes breaks down in the presence of strong oxidants. All roGFP2-based probes react similar and with fast kinetics with hypochlorous acid (HOCl) and polysulfides in vitro (*Müller et al., 2017b*; *2017a*). Addition of 10 µM or more HOCl to *E. coli* expressing any of the three probes led to instant probe oxidation (*Figure 9A–C*). As activated neutrophils are thought to produce high concentrations of HOCl through the action of myeloperoxidase (*Klebanoff, 2005*), we investigated, if this enzyme is involved in the unspecific probe oxidation in phagocytized *E. coli*. Thus, we used the

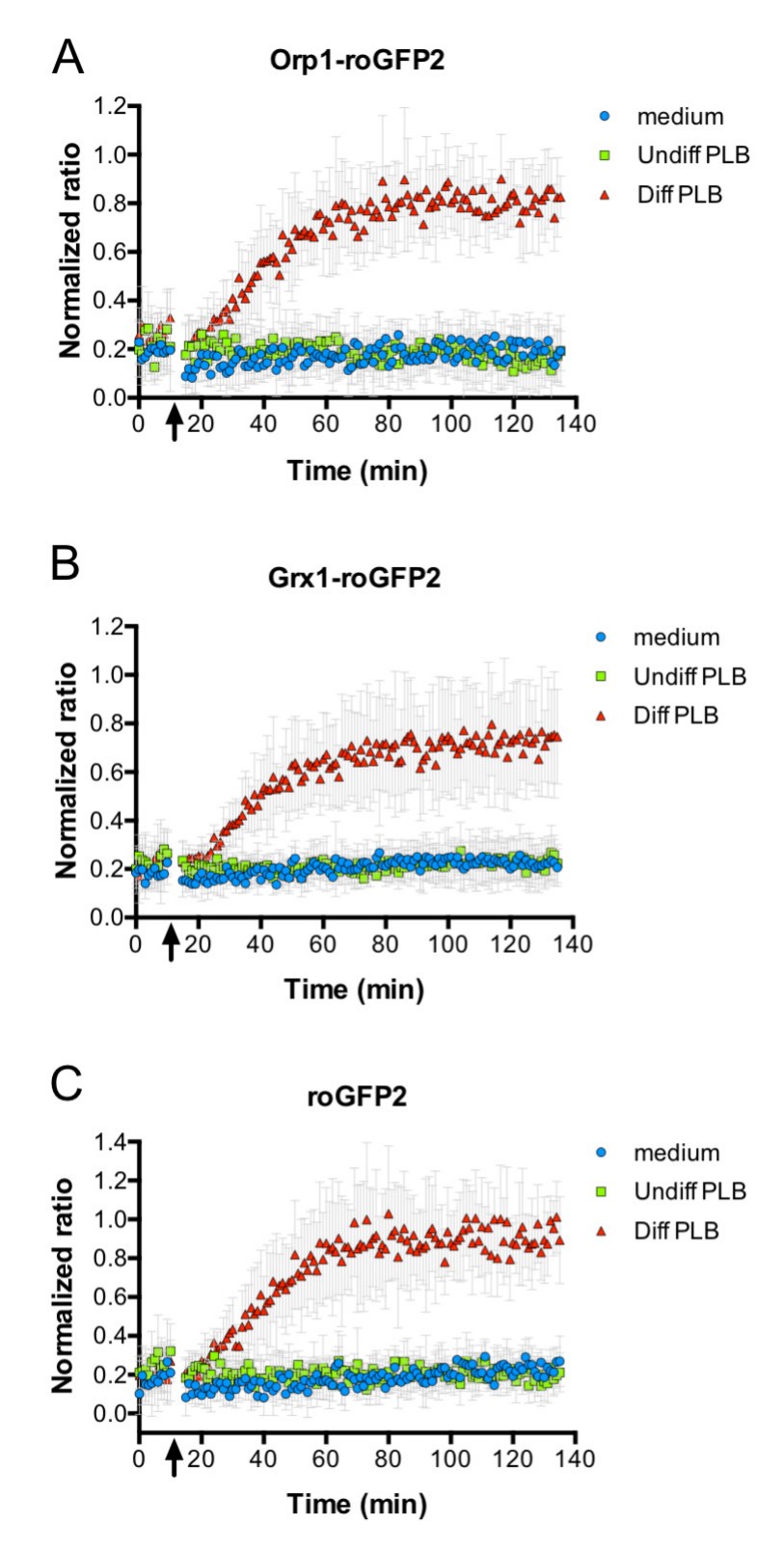

**Figure 4.** Oxidation of roGFP2-based probes expressed in *E. coli* co-cultivated with neutrophil-like cells. *E. coli* cells expressing roGFP2-Orp1 from a plasmid were incubated in a 96-well plate reader. The ratio of the fluorescence intensity at excitation wavelengths of 405 and 488 nm was calculated and plotted over time. The arrow indicates the addition of medium, undifferentiated PLB-985 cells and differentiated, neutrophil-like PLB-985. Addition of neutrophil-like cells led to probe oxidation, as evidenced by the increase in the ratio of the fluorescence intensities (**A**). The probe oxidation in

*Figure 4 continued on next page*

*Figure 4 continued*

bacteria expressing the Grx1-roGFP2 fusion probe (**B**) and unfused roGFP2 (**C**) showed kinetics virtually identical to *E. coli* cells expressing roGFP2-Orp1.

DOI: https://doi.org/10.7554/eLife.32288.040

The following source data is available for figure 4:

**Source data 1.** Numerical fluorescence plate reader data represented in *Figure 4A*.
DOI: https://doi.org/10.7554/eLife.32288.041
**Source data 2.** Numerical fluorescence plate reader data represented in *Figure 4B*.
DOI: https://doi.org/10.7554/eLife.32288.042
**Source data 3.** Numerical fluorescence plate reader data represented in *Figure 4C*.
DOI: https://doi.org/10.7554/eLife.32288.043

myeloperoxidase inhibitor 4-aminobenzoic acid hydrazide (ABAH). Pre-treatment of neutrophil-like cells with ABAH resulted in a significant attenuation of the probes' response (*Figure 10A–C*) and release of reactive species as measured with an 2',7'-Dichlordihydrofluorescein-diacetate assay (*Figure 7*). Phagocytosis was not affected by ABAH treatment (*Figure 10E–F*). We thus conclude that hypochlorous acid is the major reactive species that leads to intracellular thiol oxidation as observed in all three roGFP2-based probes.

## Discussion

Phagocytosis accompanied by the respiratory burst is an important mechanism by the innate immune system to protect against invading bacteria. However, our knowledge on the early events within the phagosome as well as inside engulfed bacteria is quite limited. This is mainly due to a lack of methods that would allow specific, spatio-temporal, and quantitative measurements of ROS and changes in the cell's redox status (*Balce and Yates, 2013*; *Kalyanaraman et al., 2012*; *Winterbourn, 2014*). To close this gap, we set up a phagocytosis assay using neutrophil-like PLB-985 cells and *E. coli* bacteria expressing roGFP2-based probes in their cytoplasm.

Our initial hypothesis was that $H_2O_2$ plays a major role in microbial oxidation during early phagocytosis, as NADPH oxidase is the first enzyme in the respiratory burst cascade and the generated superoxide is thought to be quickly converted to hydrogen peroxide (*Segal et al., 1980*; *Segal, 2005*). We therefore used roGFP2-Orp1 in *E. coli* to specifically measure $H_2O_2$ accumulation in the bacterial cell. The probe was oxidized rapidly inside *E. coli* in the presence of 1 mM $H_2O_2$, but recovery was observable within approximately 30 min. Permanent probe oxidation was achieved upon addition of 10 mM $H_2O_2$, suggesting that the bacteria are incapable of detoxifying these high concentrations.

We then tested the response of roGFP2-Orp1, but also Grx1-roGFP2 and unfused roGFP2 to co-incubation and phagocytosis by PLB-985. All probes were effectively oxidized under these conditions and showed virtually the same kinetics. This was somewhat unexpected, since the roGFP2-Orp1 probe typically shows high specificity toward hydrogen peroxide, whereas the Grx1-roGFP2 probe was designed to rapidly and specifically equilibrate with the glutathione redox couple (*Gutscher et al., 2008*): in vitro, $H_2O_2$ treatment of purified Grx1-roGFP2 did not lead to significant probe oxidation even at concentrations as high as 100 µM (*Müller et al., 2017a*; *2017b*). In the same vein, unfused roGFP2 only slowly equilibrates with the glutathione redox couple and did not show significant oxidation by hydrogen peroxide and oxidized glutathione in vivo and in vitro (*Meyer and Dick, 2010*; *Müller et al., 2017a*; *2017b*). However, we previously showed that all three probes are effectively oxidized by HOCl in vitro (*Müller et al., 2017a*; *2017b*), and HOCl is well known to be highly reactive with most thiols (*Peskin and Winterbourn, 2001*; *Storkey et al., 2014*). To test if this holds true in an in vivo setting in bacteria as well, we treated *E. coli* expressing any of the three probes with increasing concentrations of HOCl. The response was instant at all concentrations and comparable for all three probes. It is therefore likely that a complex mixture of reactive species, with HOCl as the main component, leads to the oxidation of the probes in *E. coli*.

In a parallel approach, ROS formation in differentiated PLB-985 cells stimulated with PMA was found strongly increased as well. Others have compared the activity of PLB-985 (and cell line HL-60, which is essentially the same cell line [*Drexler et al., 2003*]) to that of neutrophils. It was shown that

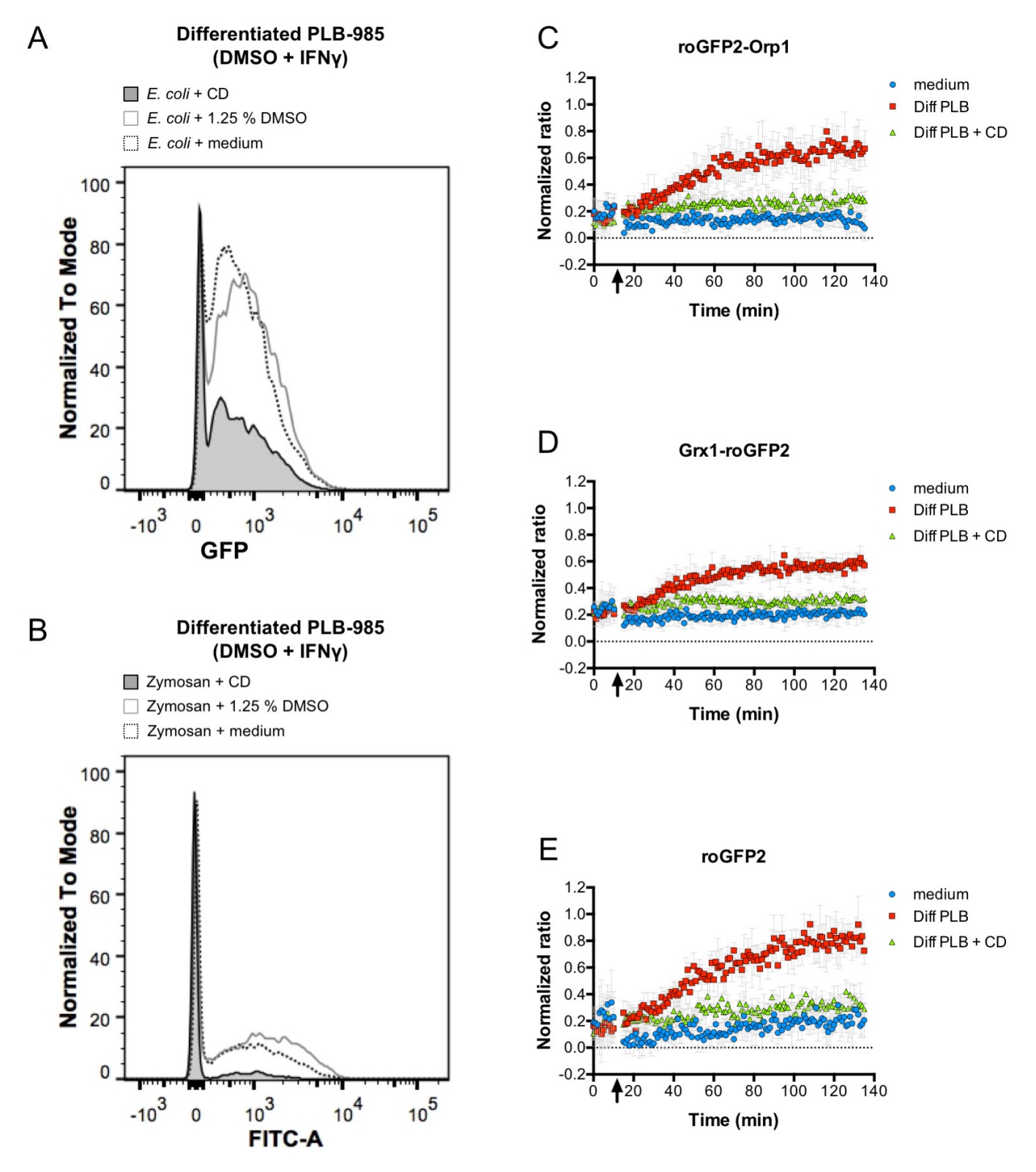

**Figure 5.** Phagocytosis is required for efficient oxidation of roGFP2-based probes. Cytochalasin D inhibits phagocytosis of *E. coli* (A) and FITC-labeled Zymosan (B) as determined by flow cytometry. Cytochalasin D (CD) treatment of differentiated PLB-985 cells inhibits oxidation of roGFP2-Orp1 expressed in *E. coli* (C). Similar findings were observed in bacteria expressing Grx1-roGFP2 (D) and unfused roGFP2 (E). Arrows indicate the addition of medium, differentiated, neutrophil-like PLB-985 cells and cytochalasin D-treated differentiated PLB-985 cells.

*Figure 5 continued on next page*

*Figure 5 continued*

DOI: https://doi.org/10.7554/eLife.32288.044

The following source data is available for figure 5:

**Source data 1.** Numerical flow cytometry data represented in *Figure 5A*, trace *E. coli* + CD.
DOI: https://doi.org/10.7554/eLife.32288.045
**Source data 2.** Numerical flow cytometry data represented in *Figure 5A*, trace *E. coli* + 1.25% DMSO.
DOI: https://doi.org/10.7554/eLife.32288.046
**Source data 3.** Numerical flow cytometry data represented in *Figure 5A*, trace *E. coli* + medium.
DOI: https://doi.org/10.7554/eLife.32288.047
**Source data 4.** Numerical flow cytometry data represented in *Figure 5B*, trace Zymosan + CD.
DOI: https://doi.org/10.7554/eLife.32288.048
**Source data 5.** Numerical flow cytometry data represented in *Figure 5B*, trace Zymosan + 1.25% DMSO.
DOI: https://doi.org/10.7554/eLife.32288.049
**Source data 6.** Numerical flow cytometry data represented in *Figure 5B*, trace Zymosan + medium.
DOI: https://doi.org/10.7554/eLife.32288.050
**Source data 7.** Numerical fluorescence plate reader data represented in *Figure 5C*.
DOI: https://doi.org/10.7554/eLife.32288.051
**Source data 8.** Numerical fluorescence plate reader data represented in *Figure 5D*.
DOI: https://doi.org/10.7554/eLife.32288.052
**Source data 9.** Numerical fluorescence plate reader data represented in *Figure 5E*.
DOI: https://doi.org/10.7554/eLife.32288.053

this cell line, when differentiated to neutrophils, expresses slightly less myeloperoxidase than blood neutrophils (*Pivot-Pajot et al., 2010*) and had a generally weaker functional response. However, in other studies, superoxide production and MPO activity have been described to be comparable or even exceeding that of human neutrophils (*Hua et al., 2000*; *Thompson et al., 1988*).

Since HOCl and other reactive species produced during the oxidative burst are ultimately derived from superoxide produced by NADPH-oxidase, roGFP2-based probe oxidation was largely diminished when PLB-985 cells lacking NOX2 activity were used in the co-cultivation assays. Our data also indicates that *E. coli* experiences the largest amount of oxidative stress within the phagolysosome, since the probe expressed in non-phagocytized bacteria was significantly less oxidized in the same co-cultivation assay (*Figure 6*; *Video 1*). This is somewhat surprising, as especially $H_2O_2$ is thought to easily diffuse through membranes (*Dupré-Crochet et al., 2013*).

Our hypothesis that HOCl could be responsible for the unspecific roGFP2 probe oxidation in phagocytized *E. coli* was further substantiated using the myeloperoxidase inhibitor ABAH. Pre-treatment of neutrophil-like cells with ABAH resulted in a significant decrease of the probes' oxidation and release of reactive species as measured with a DCF assay. Taken together, these results suggest HOCl is most likely the major reactive species responsible for roGFP2 oxidation in *E. coli* ingested by PLB-985 cells.

Our results provide some insights into the nature of ROS released within the phagosome. While production of different ROS at the onset of phagocytosis was known and described before and has been observed using roGFP2 expressed in bacteria (*van der Heijden et al., 2015*), the exact composition and concentrations of individual reactive species as well as their time-resolved release remain largely elusive (*Dupré-Crochet et al., 2013*; *Hurst, 2012*; *Klebanoff et al., 2013*; *Winterbourn and Kettle, 2013*). As soon as neutrophils ingest opsonized bacteria and the vacuole is formed, degranulation begins within seconds and NADPH oxidase becomes activated and produces superoxide, which dismutates to $H_2O_2$ (*Segal et al., 1980*; *Segal, 2005*). When the granules are fused to the phagosome, MPO levels increase, leading to the formation of HOCl and derived chloramines (*Hurst, 2012*; *Winterbourn and Kettle, 2013*). This HOCl generation in the phagolysosome could be directly observed in porcine neutrophils phagocytizing zymosan particles with the help of a chemical probe selective for HOCl (*Koide et al., 2011*). Due to its high abundance, MPO thus can be regarded as a sink for hydrogen peroxide, which would favor the view that it does not accumulate to concentrations high enough to harm bacteria (*Dupré-Crochet et al., 2013*; *Klebanoff et al., 2013*; *Winterbourn and Kettle, 2013*). Instead, HOCl and potentially HOCl-derived chloramines seem to play the major role in the oxidation of bacterial macromolecules (*Chapman et al., 2002*; *Clark and*

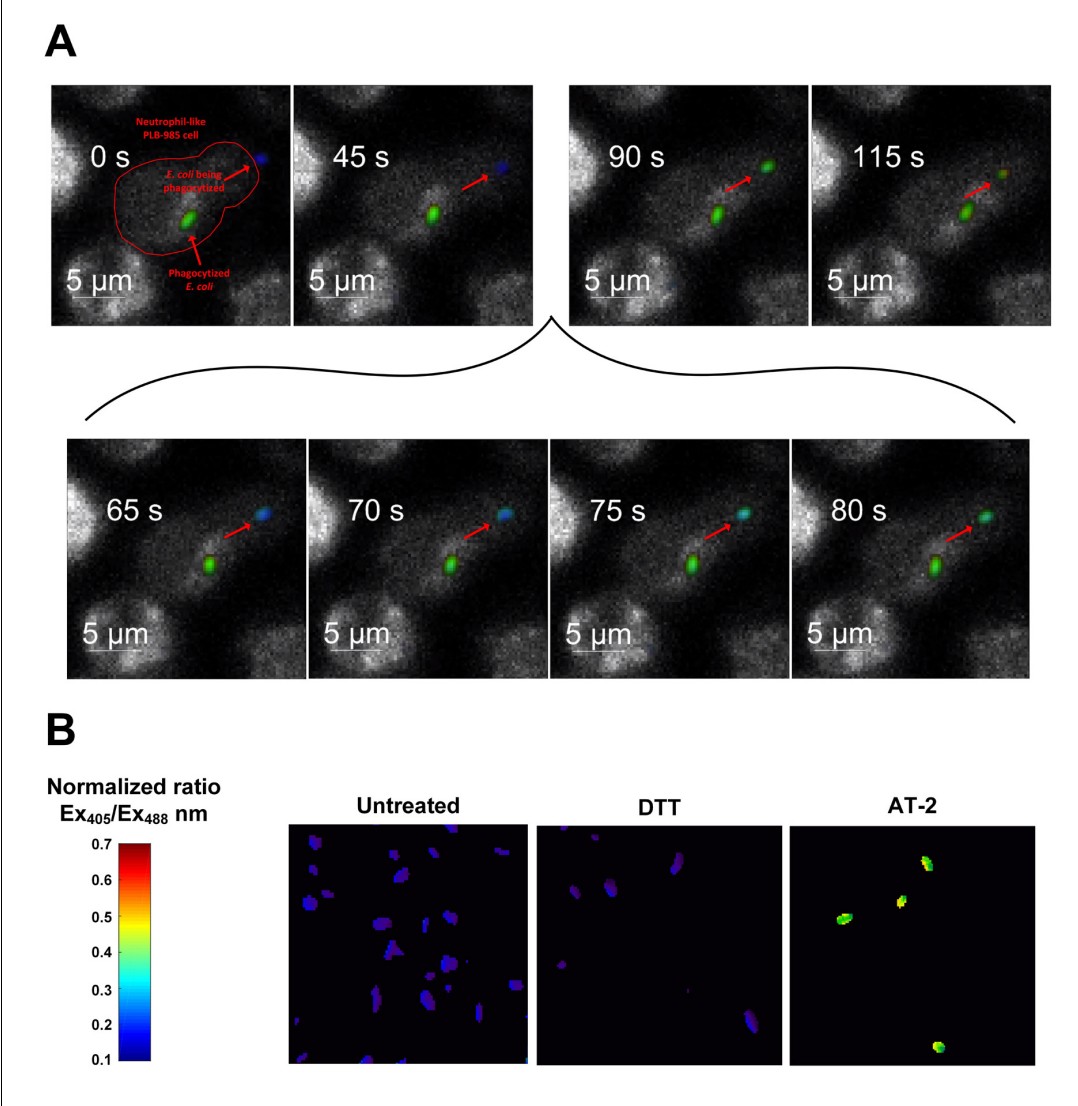

**Figure 6.** Probe oxidation occurs within seconds after phagocytosis. (**A**) Quantitative fluorescence microscopy of the redox state of roGFP2-Orp1 in *E. coli* during phagocytosis. Stills of a movie observing an individual *E. coli* cell expressing roGFP2-Orp1 (indicated by an arrow) being attacked by a neutrophil-like PLB-985 cell. The neutrophil has already phagocytized another *E. coli* cell. Upon phagocytosis, the oxidation state changes within seconds (inset 65–80 s) as illustrated based on the false color scale indicated. See also *Video 1*. (**B**) Control: *E. coli* cells in the absence of neutrophil-like cells untreated and treated with the reducing agent dithiothreitol (DTT) and the oxidizing agent Aldrithiol-2 (AT-2).
DOI: https://doi.org/10.7554/eLife.32288.054

*Borregaard, 1985*; *Klebanoff et al., 2013*; *Vissers and Winterbourn, 1987*). These results are in agreement with previous studies in which bacterial killing following phagocytosis was analyzed (*Palazzolo et al., 2005*; *Schwartz et al., 2009*). GFP bleaching in phagocytized bacteria suggested that cytoplasmic HOCl concentrations were significant (*Hurst, 2012*; *Winterbourn and Kettle, 2013*). Our approach allowed real-time tracking of the roGFP2 oxidation state during phagocytosis and thus provided strong evidence of the presence of HOCl or derived chloramines within the cytoplasm of bacteria within seconds after phagocytosis.

Interestingly, most individuals with MPO deficiency do not particularly suffer from microbial infections (*Kutter et al., 2000*; *Nauseef, 1988*; *Parry et al., 1981*). Inhibition of MPO, the enzyme producing HOCl in neutrophils, led to lower probe oxidation but did not fully inhibit oxidation, indicating that ROS produced prior to HOCl can still affect the thiol redox state of proteins in *E. coli*'s cytoplasm, or alternatively, that ABAH did not fully inhibit MPO, as has been described

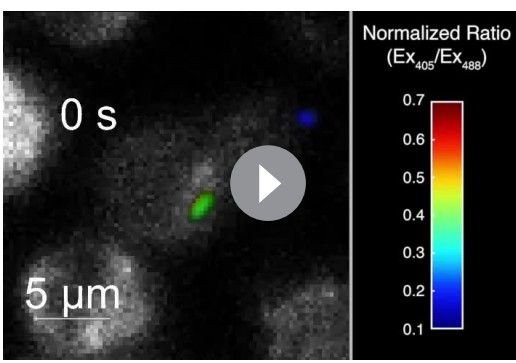

**Video 1.** Time lapsed movie of quantitative fluorescence microscopy of the redox state of roGFP2-Orp1 in *E. coli* during phagocytosis. Related to *Figure 6*. An *E. coli* cell is being attacked by a neutrophil-like PLB-985 cell. The neutrophil has already phagocytized another *E. coli* cell. Upon phagocytosis, the oxidation state changes within seconds.
DOI: https://doi.org/10.7554/eLife.32288.055

(*Björnsdottir et al., 2015*; *Parker et al., 2011*). In contrast, in the absence of NOX2, roGFP2 remained fully reduced, indicating that the absence of superoxide, which is needed for the formation of peroxynitrite, $H_2O_2$, and further derived ROS and RNS including HOCl, prevents the breakdown of the bacterial thiol redox state.

## Materials and methods

### PLB-985 culture and differentiation

The human myeloid leukemia cell line PLB-985 (obtained from DSMZ, German collection of microorganisms and cell culture) was cultured in RPMI-1640 medium supplemented with 10% FBS and 1% GlutaMAX (Life Technologies, Carlsbad, CA) at 37°C in a humidified incubator at 5% $CO_2$. Cells were authenticated based on their ability to differentiate to neutrophils and the associated expression of the respective surface markers (see below). Their mycoplasma status was not tested by us, however, all cell lines distributed by DSMZ are certified mycoplasma negative. Cell cultures were passaged twice a week to maintain a cell density between $2 \times 10^5$ and $1 \times 10^6 \cdot mL^{-1}$. For neutrophil-like phenotype differentiation, exponentially growing cells at a starting density of $2 \times 10^5 \cdot mL^{-1}$ were cultured in RPMI 1640 medium supplemented with 10% FBS, 1% GlutaMax and 1.25% DMSO for 5 days (*Pivot-Pajot et al., 2010*). The phagocytic function of the PLB-985 cells was stimulated with 2000 U $\cdot mL^{-1}$ interferon-γ (ImmunoTools, Friesoythe, Germany), added to the culture on day 4 during the differentiation period (*Tlili et al., 2011*). Cell viability was monitored by trypan blue exclusion and was typically >90%. The efficiency of differentiation was estimated by morphological analysis with May-Grünwald-Giemsa stain and flow cytometric analysis of the expression of surface markers CD11b and CD64 using specific phycoerythrin (PE)-conjugated antibodies (eBioscience, San Diego, CA). $10^5$ PE stained cells were monitored by flow cytometer BD FACSCanto II (Becton, Dickinson and Company, Franklin Lakes, NJ) equipped with three lasers, with blue (488 nm, air-cooled, 20 mW solid state), red (633 nm, 17 mW HeNe) and violet wavelengths (405 nm, 30 mW solid state). The red fluorescence (PE emission) was collected after passing through a 585/42 nm band pass (BP) filter. Data was analyzed using Flow Jo software Version 10.2 (Tree Star Inc., Ashland, OR).

### Construction of genetically encoded, redox-sensitive fluorescent probes for expression in *E. coli*

Construction of plasmid pCC_roGFP2 for expression of unfused roGFP2 (*Table 1*) in *E. coli* was described previously (*Müller et al., 2017a*). For the expression of roGFP2-Orp1 and Grx1-roGFP2 in *E. coli*, the respective gene regions were amplified using the primer pairs listed in *Table 1* from pQE-based vectors that served as template (*Table 2*). Subsequently, the PCR products were cloned into empty pCC using the restriction enzymes NdeI and EcoRI. pCC expresses proteins in *E. coli* with an IPTG-inducible Tac promoter (*Masuch et al., 2015*). *E. coli* XL1 blue was used as a cloning host. The pCC vectors containing the three redox-sensitive fluorescent probes (roGFP2, Grx1-roGFP2, and roGFP2-Orp1) were subsequently transformed into *E. coli* MG1655 and 100 µg/mL ampicillin was added to the growth medium for maintenance of the plasmid (*Table 2*).

### Fluorometric measurement of roGFP2-based probe oxidation state in *E. coli* in response to AT-2 and DTT

*E. coli* strains harboring pCC vectors containing the roGFP2-based probes (*Table 2*) were cultured in LB liquid medium with 200 µg/mL of ampicillin at 37°C overnight. The optical density at 600 nm ($OD_{600}$) was measured and the bacterial suspension was diluted to an $OD_{600}$ of 0.1 with fresh

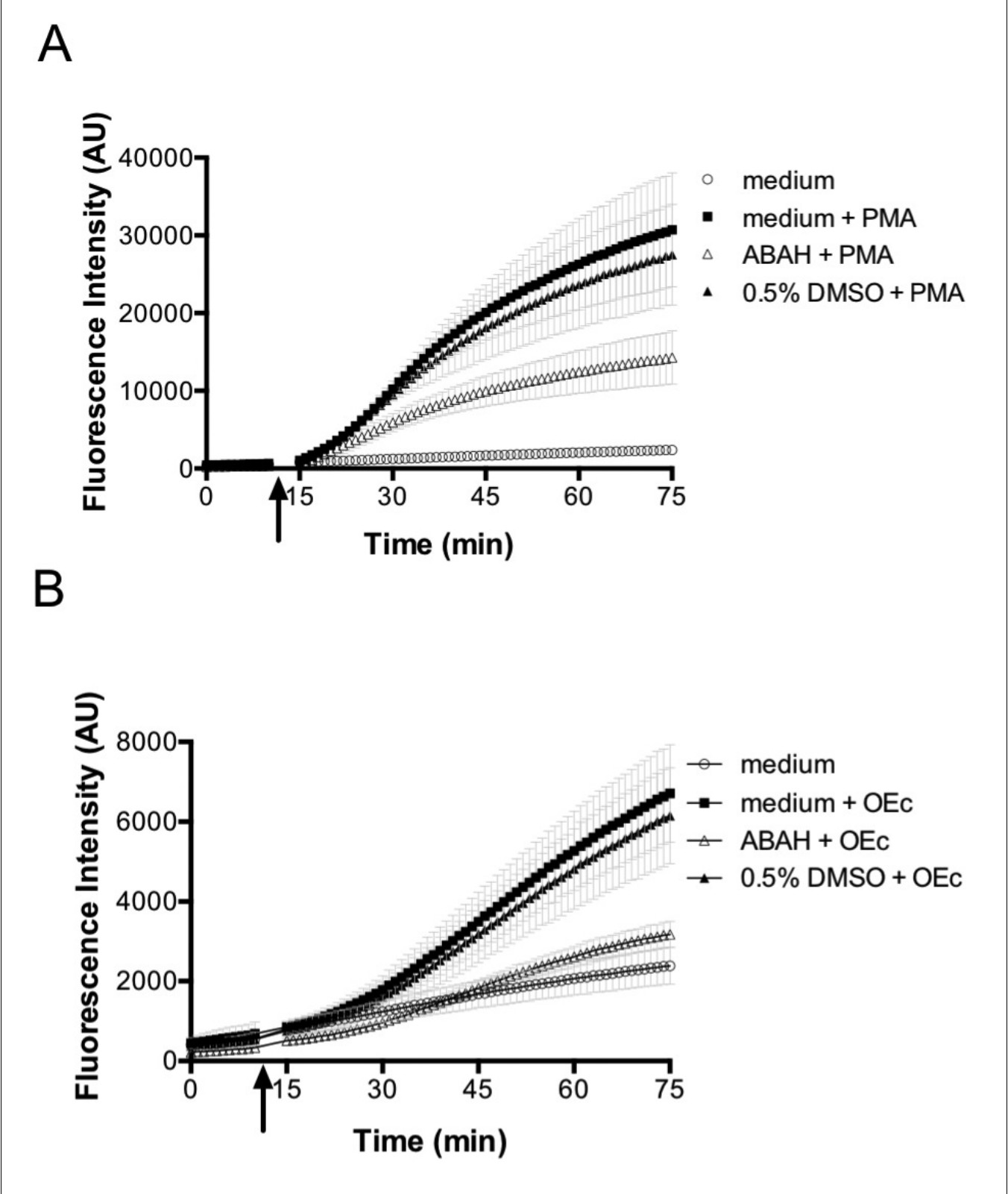

**Figure 7.** PMA and opsonized *E. coli* stimulate ROS production in differentiated neutrophil-like PLB-985 cells. Differentiated, neutrophil-like PLB-985 produce reactive species that can be detected by 2′,7′-Dichlordihydrofluorescein-diacetate (H$_2$DCFDA) oxidation when stimulated with PMA (**A**) or opsonized *E. coli* (**B**). The myeloperoxidase inhibitor ABAH (4-aminobenzoic acid hydrazide) decreases the production of reactive species. The arrow indicates the time-point of the addition of PMA or opsonized *E. coli*.
*Figure 7 continued on next page*

*Figure 7 continued*

DOI: https://doi.org/10.7554/eLife.32288.056

The following source data is available for figure 7:

**Source data 1.** Numerical fluorescence plate reader data represented in *Figure 7*.

DOI: https://doi.org/10.7554/eLife.32288.057

medium and cultured at 37°C for ~2 hr until an $OD_{600}$ of 0.5–0.8 was reached. The expression of roGFP2-based probes was then induced with 100 μM IPTG and the culture was incubated at 20°C overnight. These bacterial cells, now containing roGFP2-based probes, were then washed twice in 40 mM HEPES buffer (pH 7.4) and re-suspended in 1 mL HEPES buffer to a final $OD_{600}$ of 0.3. The fluorescence intensity was measured in an FP-8500 spectrofluorometer (Jasco, Tokyo, Japan). The emission wavelength was fixed at 510 nm and excitation wavelength was scanned from 350 to 500 nm. Bandwidths of excitation and emission were set to 5 nm. The cell suspension in the cuvette was continuously stirred with a magnetic stir bar and the temperature of the temperature controller EHC-813 (Jasco) was set to 25°C. Fluorescence excitation ratios (405/488 nm) were used as measurement of probe oxidation (*Arias-Barreiro et al., 2010*; *Meyer and Dick, 2010*). Oxidation with 100 μM aldrithiol-2 (AT-2, Sigma-Aldrich, St. Louis, MO) and reduction with 50 mM dithiothreitol (DTT, Sigma-Aldrich) were used to fully oxidize and fully reduce the probes, respectively (*Figure 1D–F*).

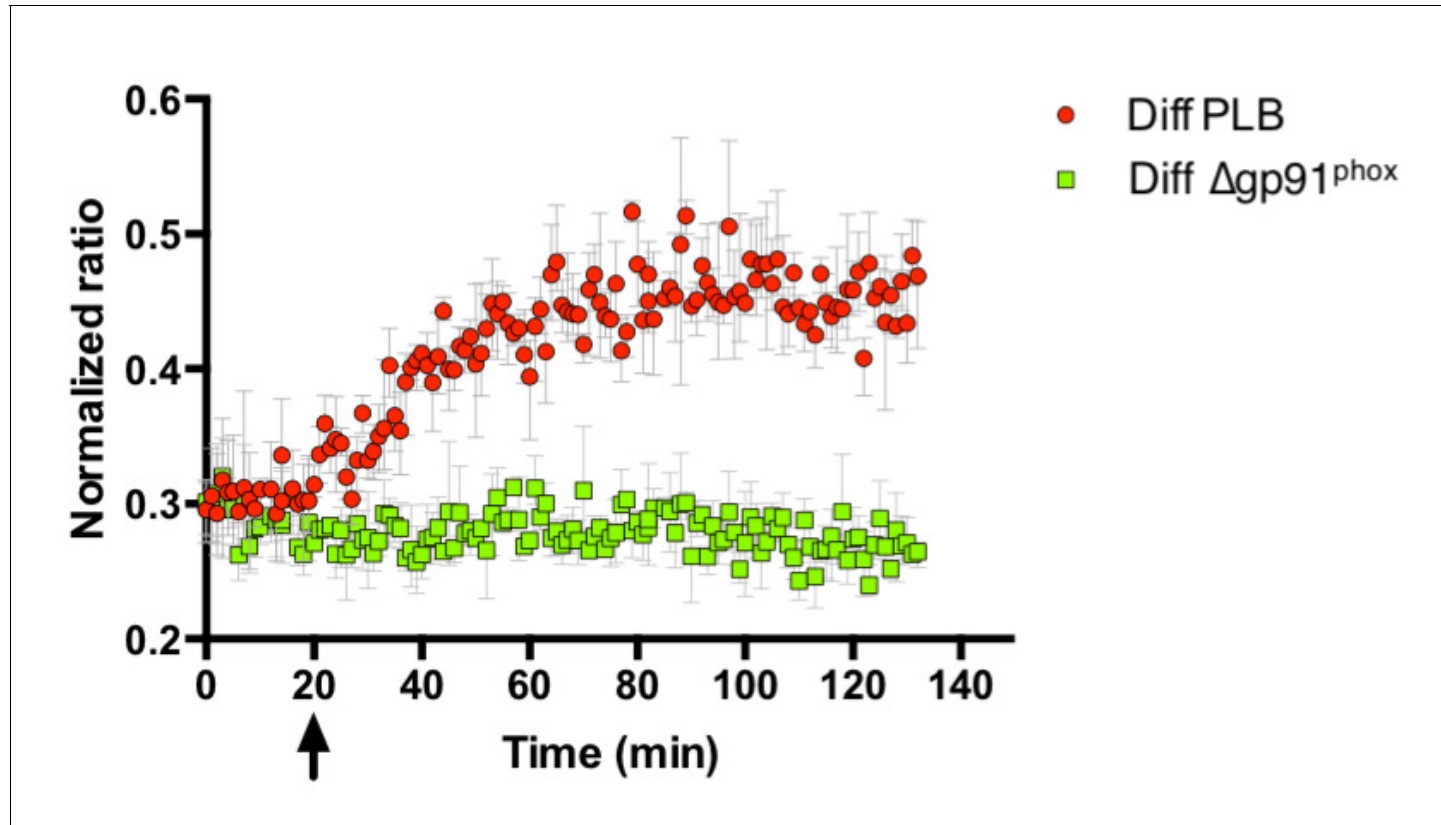

**Figure 8.** Probe oxidation in bacteria is dependent on NOX2. PLB-985 lacking gp91[phox], the catalytic subunit of NOX2 and a major intracellular producer of superoxide, no longer induce probe oxidation in *E. coli* cells expressing roGFP2-Orp1. The arrow indicates the addition of differentiated, neutrophil-like PLB-985 cells or differentiated PLB-985 cells lacking gp91[phox].

DOI: https://doi.org/10.7554/eLife.32288.058

The following source data is available for figure 8:

**Source data 1.** Numerical fluorescence plate reader data represented in *Figure 8*.

DOI: https://doi.org/10.7554/eLife.32288.059

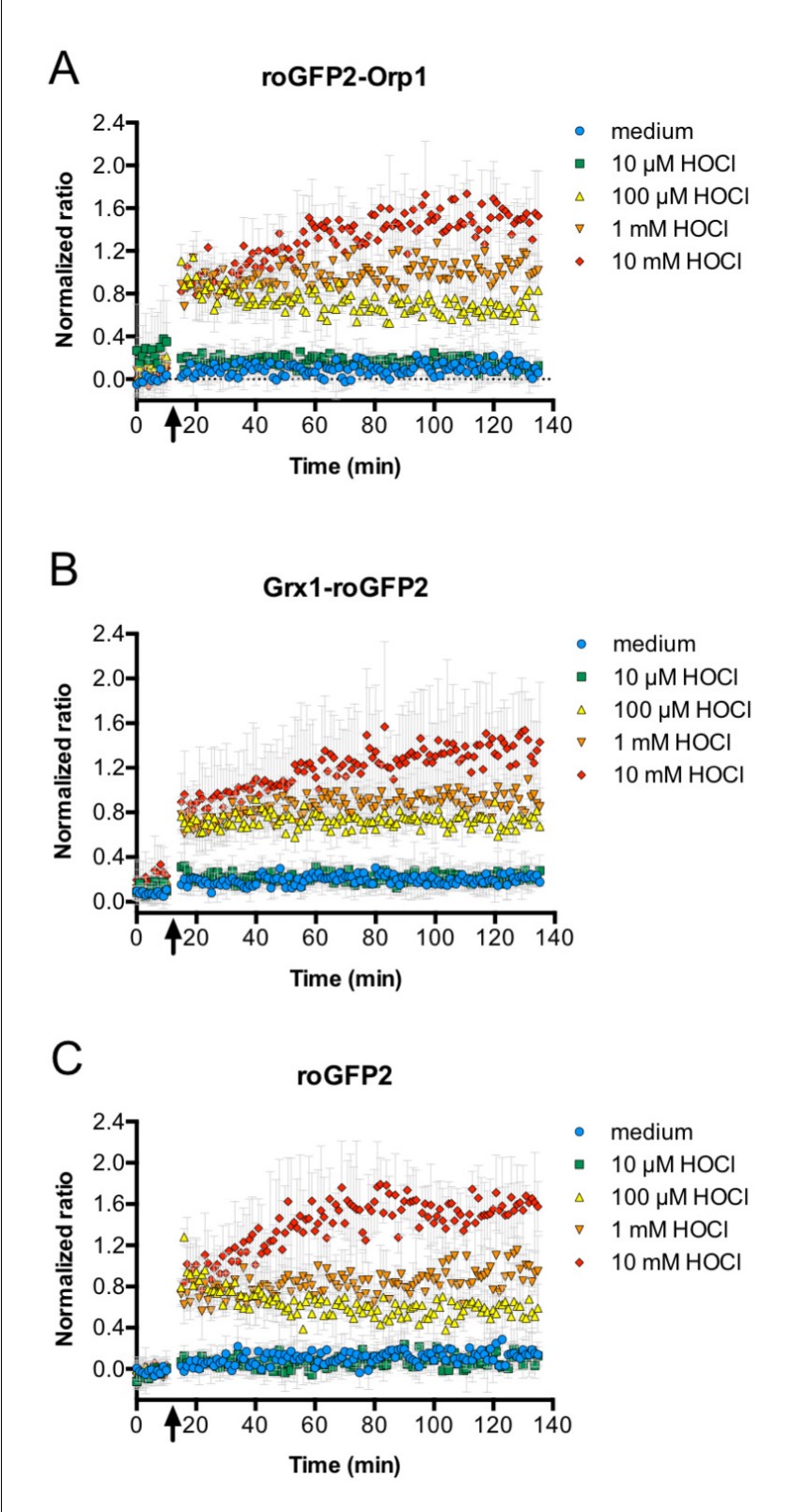

**Figure 9.** HOCl oxidizes roGFP2-based probes in *E. coli* effectively and with fast kinetics. All three roGFP2-based probes used in this study, roGFP2-Orp1 (**A**), Grx1-roGFP2 (**B**), and unfused roGFP2(**C**) are oxidized within mixing time upon addition of HOCl. *E. coli* cells expressing the respective probes were incubated in a 96-well plate reader and a time-course measurement of the ratio of the fluorescence intensity of the probe at excitation

*Figure 9 continued on next page*

*Figure 9 continued*

wavelengths of 405 and 488 nm was performed. The arrow indicates the addition of different concentrations of HOCl or medium as a control as indicated.

DOI: https://doi.org/10.7554/eLife.32288.060

The following source data is available for figure 9:

**Source data 1.** Numerical fluorescence plate reader data represented in *Figure 9A*.
DOI: https://doi.org/10.7554/eLife.32288.061
**Source data 2.** Numerical fluorescence plate reader data represented in *Figure 9B*.
DOI: https://doi.org/10.7554/eLife.32288.062
**Source data 3.** Numerical fluorescence plate reader data represented in *Figure 9C*.
DOI: https://doi.org/10.7554/eLife.32288.063

## Measurement of roGFP2-based probe oxidation state in a 96-well format

*E. coli* cells expressing roGFP2-based probes as described above were washed twice in PBS pH 7.4 and resuspended in PBS pH 7.4 with 0.5% FBS to a final $OD_{600}$ of 0.1–0.3. FBS was omitted in experiments where no PLB-985 cells were present (*Figure 1J* and *Figure 8*). Fifty microliters of this *E. coli* suspension were placed in the wells of a black, clear-bottom 96-well plate (Nunc, Rochester, NY). Fluorescence intensity was recorded during every minute for 10 min using the Synergy H1 multi-detection microplate reader (Biotek, Bad Friedrichshall, Germany) at the excitation wavelengths 405 and 488 nm and emission wavelength at 530 nm. Then 50 μL of the selected chemical solution or the respective PLB-985 cell suspension was added to the wells and the fluorescence intensity was monitored for another 2 hr under the same conditions. The signals of fully oxidized and fully reduced probes were obtained by adding 100 μM AT-2 or 50 mM DTT to the bacteria culture at the start of the 2 hr measurement. The fluorescence excitation ratios (405/488 nm) were used as measurement of probe oxidation and all values were normalized to the values obtained for fully oxidized (AT-2-treated) and for fully reduced (DTT-treated) bacteria cultures.

## Phagocytosis of bacteria by neutrophil-like PLB-985 cells

*E. coli* cells expressing roGFP-based probes (as described above) were opsonized by incubation for 30 min at 37°C with 1 mg. $mL^{-1}$ human immunoglobulin G (Sigma-Aldrich) in PBS pH 7.4, then washed twice in PBS and re-suspended in PBS with 0.5% FBS to an $OD_{600}$ of 0.1. 50 μL of opsonized *E. coli* was then added to 50 μL of differentiated PLB-985 cell suspension ($10^7$ $mL^{-1}$) in PBS with 0.5% FBS. Thus a ratio of 10 *E. coli* bacteria per PLB-985 cell was used to initiate phagocytosis. Cells and bacteria were then co-incubated at 37°C, and phagocytosis was stopped by adding 100 μL of ice-cold PBS at certain time points. Cells were fixed with 4% paraformaldehyde and samples were kept on ice until subjected to flow cytometry. Samples were analyzed using a BD FACSCanto II flow cytometer (Becton, Dickinson and Company) with an argon laser operating at 488 nm using the 530/30 emission filter to detect the fluorescence of the roGFP2-based probes expressed in phagocytized bacteria. For each sample, a total of 10,000 viable cells were counted. The mean fluorescence intensity (MFI) multiplied by the percentage of viable cells that had ingested fluorescent particles was used to evaluate the phagocytic capacity of PLB-985 cells. In some experiments, phagocytosis was inhibited by treatment of differentiated PLB-985 cells with 250 μM cytochalasin D (Sigma-Aldrich) for 30 min at 37°C before co-incubation with opsonized bacteria.

## Fluorescence microscopy

Paraformaldehyde-fixed cells from the phagocytosis assay were centrifuged at 200 *g* for 5 min and resuspended in PBS containing 0.1% Tween 20 and 2.5 μg. $mL^{-1}$ TRITC-conjugated phalloidin (Sigma-Aldrich). Cells were incubated with phalloidin for 40 min at room temperature, protected from light. Excess phalloidin was removed by centrifugation and cells were resuspended by pipetting up and down in buffered glycerol containing 0.5 μg/mL DAPI (Sigma-Aldrich). Cells were then visualized on a slide in a BX51 fluorescence microscope with a U-UCD8 condenser, a U-LH100HGAPO burner, a U-RFL-T power supply, a 63X/1.4 NA oil objective and a 450–490 nm excitation/500–550 emission bandpass filter (Olympus, Tokyo, Japan). Images were collected with a CC12 digital color

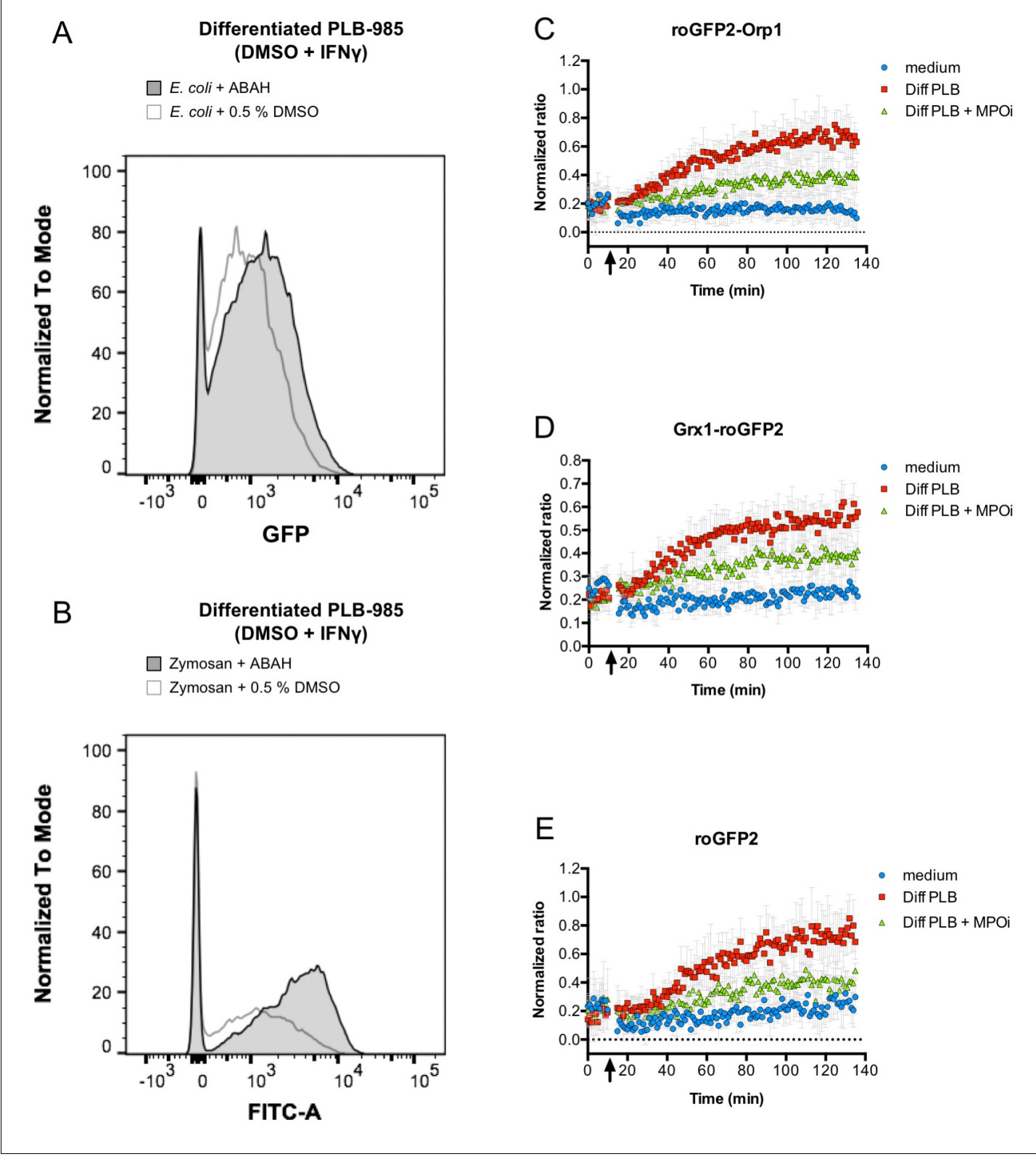

**Figure 10.** Inhibition of myeloperoxidase abrogates probe oxidation. Addition of myeloperoxidase inhibitor 4-aminobenzoic acid hydrazide (ABAH) does not inhibit phagocytosis of opsonized *E. coli* (**A**) or Zymosan (**B**), as evidenced by flow cytometry. However, neutrophil-like cells pre-incubated with ABAH are less capable of oxidizing roGFP2-Orp1 (**C**), Grx1-roGFP2 (**D**), or unfused roGFP2 expressed in *E. coli* (**E**). Arrows indicate the addition of PLB 985 cells or medium to bacteria.

*Figure 10 continued on next page*

*Figure 10 continued*

DOI: https://doi.org/10.7554/eLife.32288.064

The following source data is available for figure 10:

**Source data 1.** Numerical flow cytometry data represented in *Figure 10A*, trace *E. coli* + ABAH.
DOI: https://doi.org/10.7554/eLife.32288.065
**Source data 2.** Numerical flow cytometry data represented in *Figure 10A*, trace *E. coli* + 0.5% DMSO.
DOI: https://doi.org/10.7554/eLife.32288.066
**Source data 3.** Numerical flow cytometry data represented in *Figure 10A*, trace Zymosan + ABAH.
DOI: https://doi.org/10.7554/eLife.32288.067
**Source data 4.** Numerical flow cytometry data represented in *Figure 10A*, trace Zymosan + 0.5% DMSO.
DOI: https://doi.org/10.7554/eLife.32288.068
**Source data 5.** Numerical fluorescence plate reader data represented in *Figure 10C*.
DOI: https://doi.org/10.7554/eLife.32288.069
**Source data 6.** Numerical fluorescence plate reader data represented in *Figure 10D*.
DOI: https://doi.org/10.7554/eLife.32288.070
**Source data 7.** Numerical fluorescence plate reader data represented in *Figure 10E*.
DOI: https://doi.org/10.7554/eLife.32288.071

camera and the Cell Imaging Software (Olympus) and composite figures were prepared using ImageJ (*Schneider et al., 2012*) and Photoshop CS5 (Adobe Systems, San Jose, CA) software.

## Fluorescence live-cell imaging and ratiometric image processing

Differentiated PLB-958 cells were stained with 0.25 µM Celltracker Deep Red (Thermo Fisher Scientific, USA) in RPMI 1680 at 37°C and 5% $CO_2$ for 30 min, washed once with PBS and diluted in PBS with 0.5% FBS to a final concentration of $10^7$ cells. $mL^{-1}$. 1 mL of the cell suspension was poured onto an imaging dish (µ-Dish 35 mm, high, Ibidi, DE). Opsonized roGFP2-Orp1 *E. coli* cells were added with a ratio of five *E. coli* cells to one PLB-958 cell. Fluorescence images were acquired with an LSM 880 ELYRA PS.1 microscope (Carl Zeiss Microscopy GmbH, Jena, Germany). Images were acquired in three different channels: channel I: $Ex_{405nm}/Em_{513nm}$, channel II: $Ex_{488nm}/Em_{513nm}$, channel III: $Ex_{561nm}/Em_{674nm}$, bandwidth settings channels I and II: 13 nm, channel III: 59 nm. Individual single channel images were exported using ZEN 2.1 (Zeiss, DE). Ratiometric images were generated with ImageJ 1.51e (National Institutes of Health, USA) as described (*Collins, 2007*). The image background was corrected using a rolling ball algorithm and images were transferred to 32-bit format. Images were thresholded and converted to binary mask. Images from channel I and channel II were aligned using the ImageJ plugin „MultiStackReg" and ratiometric images were calculated using „Ratio Plus". Ratiometric images were colored using the „Lookup table" feature from the plugin „NucMed". The display range of all ratiometric images were adjusted to the same range before converting them into RGB format. For the ratiometric time series assembly, images were smoothed in order to reduce noise. After background subtraction, normalized 405/488 nm ratio image series were calculated and assembled to a movie using Software kindly provided by *Fricker (2016)*.

## DCFH oxidation assay

Intracellular oxidation of 2',7'-dichlorodihydrofluorescein (DCFH) to 2',7'-dichlorofluorescein (DCF) by PLB-985 cells was analyzed using the cell permeable derivative 2',7'-dichlorodihydrofluorescein

**Table 1.** Primers.

| Primer name | Sequence |
| --- | --- |
| Orp1-Fw | cccccccatatggtgagcaagggcgagga |
| Orp1-Rv | gggggggaattcttattccacctctttcaa |
| Grx-Fw | cccccccatatggctcaagagtttgtgaac |
| Grx-Rv | gggggggaattcttacttgtacagctcgtc |

DOI: https://doi.org/10.7554/eLife.32288.072

**Table 2.** Bacterial strains and plasmids.

| Strain or plasmid | Relevant genotype or description | Source or reference |
|---|---|---|
| *Strains* | | |
| *E. coli* XL1 blue | *rec*A1 *end*A1 *gyr*A96 *thi*-1 *hsd*R17 *sup*E44 *rel*A1 *lac* | Stratagene |
| *E. coli* MG1655 | F- lambda- *ilvG*- *rfb*50 *rph*-1 | (*Blattner et al., 1997*) |
| E. coli AM39 | MG1655 pCC_roGFP2-orp1 | This work |
| E. coli AM180 | MG1655 pCC_grx1-roGFP2 | This work |
| E. coli AM181 | MG1655 pCC_roGFP2 | This work |
| *Plasmids* | | |
| pCC | TAC-MAT-Tag-2 derivative; p*tac* | (*Masuch et al., 2015*) |
| pQE60_roGFP2-orp1-his-QC | pQE60 carrying *roGFP2-orp1*-his$_6$; removed EcoRI site; pT5-*lac* promoter | (*Müller et al., 2017a*) |
| pQE60_grx1_roGFP2-his-QC | pQE60 carrying grx1-*roGFP2*-his$_6$: removed EcoRI site; pT5-*lac* | (*Müller et al., 2017a*) |
| pCC_roGFP2 | *roGFP2*; p*tac* | (*Müller et al., 2017a*) |
| pCC_roGFP2-orp1 | *roGFP2-orp1*; p*tac* | This work |
| pCC_grx1-roGFP2 | *grx1-roGFP2*; p*tac* | This work |

DOI: https://doi.org/10.7554/eLife.32288.073

diacetate (DCFH-DA). PLB-985 cells ($10^{7.}$ mL$^{-1}$) were pre-incubated with 1.25 μM DCFH-DA (Thermo Fisher Scientific, Waltham, MA) in PBS with 0.5% FBS during 15 min at 37°C. Then 50 μL were placed in the wells of a black, clear-bottom 96-well plate (Nunc). Fluorescence intensity was recorded every 1 min for 10 min using the Synergy H1 multi-detection microplate reader (Biotek) at an excitation wavelength of 488 nm and an emission wavelength of 525 nm. Cells were then activated by addition of 50 ng · mL$^{-1}$ phorbol 12-myristate 13-acetate (PMA; Sigma-Aldrich) or incubation with 10-fold excess of *E. coli* bacteria. Fluorescence intensity was then recorded every 1 min for 1 hr.

## Myeloperoxidase inhibition

Myeloperoxidase activity in neutrophil-like PLB-985 cells was inhibited by pre-incubation of cells with 500 μM 4-aminobenzoic acid hydrazide (ABAH; Sigma-Aldrich) for 30 min at 37°C prior to subsequent experiments.

## Data analysis

All 96-well-plate-based fluorescence measurement experiments were repeated at least three times with biologically independent replicates and the results are expressed as the mean ±standard deviation as represented by error bars. Representative data is shown for fluorescence spectroscopy, flow cytometry and microscopy image data.

## Acknowledgements

We thank Dr. Oliver Nüsse for the generous donation of cell line PLB-985 Δgp91[phox]. We thank Dairovys Jimenez for assistance in the establishment of this cell line in our lab. We are grateful to Dr. Mark Fricker for his help in analyzing the fluorescence microscopy data and his invaluable assistance in preparing the supplemental movie. We thank Dr. Markus Schwarzländer for fruitful discussion of our data. AD acknowledges support from CNPq (Ciência sem Fronteiras postdoctoral fellowship PDE 248836/2013–7). Principal funding was provided in the context of priority program 1710 'Dynamics of Thiol-based Redox Switches in Cellular Physiology' of the German Research Foundation (DFG) through grants ME1567/9-2 to AJM and LE2905/1-2 to LIL and through DFG grant INST 213/840-1 FUGG to KFW.

## Additional information

### Funding

| Funder | Grant reference number | Author |
|---|---|---|
| Conselho Nacional de Desenvolvimento Científico e Tecnológico | PDE 248836/2013-7 | Adriana Degrossoli |
| Deutsche Forschungsgemeinschaft | INST 213/840-1 FUGG | Konstanze F Winklhofer |
| Deutsche Forschungsgemeinschaft | ME1567/9-2 | Andreas J Meyer |
| Deutsche Forschungsgemeinschaft | LE2905/1-2 | Lars I Leichert |

The funders had no role in study design, data collection and interpretation, or the decision to submit the work for publication.

### Author contributions

Adriana Degrossoli, Conceptualization, Investigation, Visualization, Writing—original draft; Alexandra Müller, Investigation, Writing—review and editing; Kaibo Xie, Verian Bader, Investigation, Visualization; Jannis F Schneider, Investigation; Konstanze F Winklhofer, Supervision, Funding acquisition; Andreas J Meyer, Software, Methodology; Lars I Leichert, Conceptualization, Supervision, Funding acquisition, Visualization, Project administration, Writing—review and editing

### Author ORCIDs

Konstanze F Winklhofer (iD) http://orcid.org/0000-0002-7256-8231
Andreas J Meyer (iD) https://orcid.org/0000-0001-8144-4364
Lars I Leichert (iD) http://orcid.org/0000-0002-5666-9681

### Decision letter and Author response

Decision letter https://doi.org/10.7554/eLife.32288.076
Author response https://doi.org/10.7554/eLife.32288.077

## Additional files

### Supplementary files

• Transparent reporting form
DOI: https://doi.org/10.7554/eLife.32288.074

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
