## [Decision Letter]

Thank you for submitting your article "Neutrophil-generated HOCl is the major factor in the breakdown of the thiol redox potential in phagocytized bacteria" for consideration by *eLife*. Your article has been reviewed by three peer reviewers, and the evaluation has been overseen by a Reviewing Editor and Gisela Storz as the Senior Editor. The following individuals involved in review of your submission have agreed to reveal their identity: Clare Hawkins (Reviewer #2).

The reviewers have discussed the reviews with one another and the Reviewing Editor has drafted this decision to help you prepare a revised submission.

Summary:

This is a well-performed study that presents novel and interesting results. The authors show that roGFP proteins expressed in *E. coli* are sensitive to oxidation by hypochlorous acid (or derived oxidants), that there is myeloperoxidase-dependent oxidation of the probe when expressing bacteria are ingested by neutrophils, and that the probe is capable of real-time monitoring of HOCl production in phagocytosing neutrophils. The latter observation has potential for wide application.

Essential revisions:

There is a major issue with the context in which the results are described, i.e that the authors expected to see specificity of the probes that have been developed to be sensitive to H2O2 or glutathione redox state, then express surprise that this was not the case. To the reviewers this is not at all surprising, however. It is well known that HOCl is highly reactive with almost all thiols (see for example Peskin, 2001; Storkey, 2014) and it would be expected that all the thiol-based probes would be prime targets. Furthermore, the authors themselves have just published a paper (Mueller et al., 2017) showing non-selectivity of the probes for HOCl. The manuscript should be revised and written from the perspective of this prior knowledge. It is fine to present the data showing no difference between the probes with either the isolated or phagocytosed bacteria, but the results should be described and interpreted on the basis of previous knowledge and the potential value of roGFP-expressing cells for following events after phagocytosis. The Discussion section in particular should be revised significantly.

1) The tone of the article is to present the observations in terms of sensing of HOCl rather than considering them more simply as a reflection of widespread thiol oxidation. For example, "thiol redox potential" in the title, abstract and elsewhere, and "the bacteria sense redox stress" (subsection “Probe oxidation is dependent on effective phagocytosis”) imply a functional effect. It is not clear from the data presented, whether oxidation of the probe is influenced by the bacterial redox environment (as implied in the Title) or reflecting a direct reaction of the probes. The authors need to change their wording. Sensing may be involved, but we do not yet know. It may become more apparent after further work establishing when oxidation occurs – before or after the bacteria are killed.

2) Please clarify what medium was used to treat the bacteria with HOCl (e.g. for Figure 8), and whether it may have influenced the response. Subsection “Fluorometric measurement of roGFP2-based probe oxidation state in *E. coli”* says they were in HEPES, subsection “Measurement of roGFP2-based probe oxidation state in a 96-well format” in PBS with FBS, and it is not always clear which were used for which experiments. It is important to note that HOCl reacts rapidly with HEPES to form a chloramine, which is likely to mediate much of the effect of HOCl. Also, FBS would scavenge some of the HOCl and decrease sensitivity to low concentrations.

3) Figure 7. This gives qualitative information that the PLB-985 cells undergo an oxidative burst. However, it would be helpful to have some quantitative information, for example with cytochrome c, to show how active they are compared with neutrophils, and also to know relative MPO levels. This information may be available in the literature, otherwise it would be good to include experimental values. Also, it would be worth looking more closely at the fluorescence spectra, as previous studies of oxidant formation in the phagosome show that HOCl forms a product with a slightly shifted fluorescence spectrum compared to DCF, e.g. Tlili et al., 2011.

4) Figure 9, subsection “Probe oxidation 290 is dependent on myeloperoxidase” andthe Discussion section. Inhibition by ABAH indicates a role for MPO. However, on the basis of the evidence presented, it cannot be concluded that partial inhibition indicates a contribution by other oxidant(s). An alternative is that inhibition by ABAH is not 100% effective. In fact, the latter explanation is more likely, as others have shown only partial inhibition of intracellular MPO by ABAH (see comment in Parker, 2011; Bjornsdottir, 2015). The conclusion of the last paragraph of the Discussion section should be changed accordingly.

5) In revising the Discussion section please provide less repetition of the results, and give more critical analysis and comparison with other studies that have attempted to quantify the composition, concentration and kinetics of oxidant formation (e.g. Koide et al., 2011 – use of HOCl^-^specific probe and van der Heijen et al., 2015 – cited in the Introduction).

---

## [Author Response]

Essential revisions:There is a major issue with the context in which the results are described, i.e that the authors expected to see specificity of the probes that have been developed to be sensitive to H2O2 or glutathione redox state, then express surprise that this was not the case. To the reviewers this is not at all surprising, however. It is well known that HOCl is highly reactive with almost all thiols (see for example Peskin, 2001; Storkey, 2014) and it would be expected that all the thiol-based probes would be prime targets. Furthermore, the authors themselves have just published a paper (Mueller et al., 2017) showing non-selectivity of the probes for HOCl. The manuscript should be revised and written from the perspective of this prior knowledge. It is fine to present the data showing no difference between the probes with either the isolated or phagocytosed bacteria, but the results should be described and interpreted on the basis of previous knowledge and the potential value of roGFP-expressing cells for following events after phagocytosis. The Discussion section in particular should be revised significantly.

We understand the reviewers concerns and we have now incorporated the results described in the former chapter “Both the Grx1-roGFP2 fusion probe and unfused roGFP2 show oxidation kinetics identical to roGFP2-Orp1 in *E. coli* exposed to neutrophils” into the chapters describing the respective roGFP2-Orp1 results. We have thus re-written the manuscript from the perspective of our prior knowledge regarding the probes’ in vitro-specificity. However, since it could be a common (although, as the reviewers rightly point out, naïve) assumption that these probes are specific even in the highly oxidizing environment of the phagolysosme, we still state that initial hypothesis briefly in the discussion to dismiss it based on our data. Probe specificity was indeed our initial assumption and only the results obtained in our study led us to perform the above mentioned in vitro-characterization of the roGFP2-based probes.

1) The tone of the article is to present the observations in terms of sensing of HOCl rather than considering them more simply as a reflection of widespread thiol oxidation. For example, "thiol redox potential" in the title, abstract and elsewhere, and "the bacteria sense redox stress" (subsection “Probe oxidation is dependent on effective phagocytosis”) imply a functional effect. It is not clear from the data presented, whether oxidation of the probe is influenced by the bacterial redox environment (as implied in the Title) or reflecting a direct reaction of the probes. The authors need to change their wording. Sensing may be involved, but we do not yet know. It may become more apparent after further work establishing when oxidation occurs – before or after the bacteria are killed.

As background for the reviewers: Our preliminary redox proteomic work suggests that there is indeed a breakdown of the thiol redox potential in the sense that we see a dramatic shift towards more oxidized thiols in the majority of cytoplasmic proteins in phagocytized bacteria vs. non-phagocytized bacteria. Since we have not yet published these results, we changed the tone of the article and no longer refer to redox sensing and breakdown of the redox potential. Specifically, we have changed the Title to “Neutrophil-generated HOCl leads to non-specific thiol oxidation in phagocytized bacteria”, changed “thiol redox potential” accordingly, where appropriate, and changed the offending “sense” to “experience”.

2) Please clarify what medium was used to treat the bacteria with HOCl (e.g. for Figure 8), and whether it may have influenced the response. Subsection “Fluorometric measurement of roGFP2-based probe oxidation state in E. coli” says they were in HEPES, subsection “Measurement of roGFP2-based probe oxidation state in a 96-well format” in PBS with FBS, and it is not always clear which were used for which experiments. It is important to note that HOCl reacts rapidly with HEPES to form a chloramine, which is likely to mediate much of the effect of HOCl. Also, FBS would scavenge some of the HOCl and decrease sensitivity to low concentrations.

Only the experiments displayed in Figure 1 (which did not use HOCl) were performed in HEPES buffer. We now clarify this in the header and some of the wording of the respective Materials and methods section (subsections “DCFH oxidation assay”,” Data analysis”.

We now also clarify that FBS was only used in experiments involving PLB-985 cells (Acknowledgements section). Since the bacterial cells presumably experienced the redox stress upon phagocytosis, the scavenging activity of 0.5% FBS is probably negligible in comparison to the inherent scavenging capacity of the contents of the phagolysosomal compartment.

3) Figure 7. This gives qualitative information that the PLB-985 cells undergo an oxidative burst. However, it would be helpful to have some quantitative information, for example with cytochrome c, to show how active they are compared with neutrophils, and also to know relative MPO levels. This information may be available in the literature, otherwise it would be good to include experimental values. Also, it would be worth looking more closely at the fluorescence spectra, as previous studies of oxidant formation in the phagosome show that HOCl forms a product with a slightly shifted fluorescence spectrum compared to DCF, e.g. Tlili et al., 2011.

We now discuss in detail the MPO and NOX2 activity in PLB-985 cells in comparison to primary neutrophils (subsection “Construction of genetically encoded, redox-sensitive fluorescent probes for expression in *E. coli”*). In our plate-based assays, however, we were not able to measure a full spectrum with enough precision to measure the relatively subtle shift described by Tlili *et al.* We thus cannot draw a conclusion as to the exact nature of the reactive species leading to fluorogenic probe oxidation.

4) Figure 9, subsection “Probe oxidation 290 is dependent on myeloperoxidase” andthe Discussion section. Inhibition by ABAH indicates a role for MPO. However, on the basis of the evidence presented, it cannot be concluded that partial inhibition indicates a contribution by other oxidant(s). An alternative is that inhibition by ABAH is not 100% effective. In fact, the latter explanation is more likely, as others have shown only partial inhibition of intracellular MPO by ABAH (see comment in Parker, 2011; Bjornsdottir, 2015). The conclusion of the last paragraph of the Discussion section should be changed accordingly.

We thank the reviewers for the comment. We now acknowledge this fact and discuss it accordingly (subsection “Phagocytosis of bacteria by neutrophil-like PLB-985 cells”).

5) In revising the Discussion section please provide less repetition of the results, and give more critical analysis and comparison with other studies that have attempted to quantify the composition, concentration and kinetics of oxidant formation (e.g. Koide et al., 2011 – use of HOCl^-^specific probe and van der Heijen et al., 2015 – cited in the Introduction).

While revising the Discussion section, in regards to the above mentioned concerns, we have also significantly shortened the Discussion section to avoid redundancy with the Results section. We now discuss our work in the context of the above-mentioned studies.